# Mecp2 fine-tunes quiescence exit by targeting nuclear receptors

Jun Yang[1†], Shitian Zou[2†], Zeyou Qiu[2†], Mingqiang Lai[2†], Qing Long[3], Huan Chen[2], Ping lin Lai[1], Sheng Zhang[2], Zhi Rao[1], Xiaoling Xie[2], Yan Gong[2], Anling Liu[3*], Mangmang Li[2*], Xiaochun Bai[1,2*]

[1]Guangdong Provincial Key Laboratory of Bone and Joint Degenerative Diseases, The Third Affiliated Hospital of Southern Medical University, Guangzhou, China; [2]State Key Laboratory of Organ Failure Research, Department of Cell Biology, School of Basic Medical Sciences, Southern Medical University, Guangzhou, China; [3]Department of Biochemistry, School of Basic Medical Sciences, Southern Medical University, Guangzhou, China

**Abstract** Quiescence (G0) maintenance and exit are crucial for tissue homeostasis and regeneration in mammals. Here, we show that methyl-CpG binding protein 2 (Mecp2) expression is cell cycle-dependent and negatively regulates quiescence exit in cultured cells and in an injury-induced liver regeneration mouse model. Specifically, acute reduction of Mecp2 is required for efficient quiescence exit as deletion of Mecp2 accelerates, while overexpression of Mecp2 delays quiescence exit, and forced expression of Mecp2 after Mecp2 conditional knockout rescues cell cycle reentry. The E3 ligase Nedd4 mediates the ubiquitination and degradation of Mecp2, and thus facilitates quiescence exit. A genome-wide study uncovered the dual role of Mecp2 in preventing quiescence exit by transcriptionally activating metabolic genes while repressing proliferation-associated genes. Particularly disruption of two nuclear receptors, *Rara* or *Nr1h3*, accelerates quiescence exit, mimicking the Mecp2 depletion phenotype. Our studies unravel a previously unrecognized role for Mecp2 as an essential regulator of quiescence exit and tissue regeneration.

## eLife assessment

This **fundamental** study provides insights into the mechanism controlling cell cycle reentry, establishing a regulatory role for Mecp2 degradation in shifting transcription from metabolic to proliferation genes during quiescence exit. The evidence, which includes experimental data from in vitro cell culture and an in vivo injury-induced liver regeneration model, is **convincing** but the trigger for MeCP2 degradation and how MeCP2 differentially regulates proliferation and metabolic genes remains unclear.

## Introduction

Cellular quiescence, also referred to as G0, is a reversible non-proliferating state. Quiescent cells can be reactivated to exit from the quiescent state and reenter the actively cycling states (G1, S, G2, and M phases) in response to certain intrinsic or extrinsic signals. Increasing evidence indicates that quiescence is not a passive non-proliferating state but is rather an active metabolic state maintained by certain transcriptional programs (*Roche et al., 2017*; *Yao, 2014*). The switch from quiescence to proliferation is coupled with extensive changes in the transcriptional program coordinating metabolic and proliferation dynamics. The tightly orchestrated quiescence exit is crucial for tissue homeostasis and regeneration after injury, especially in the liver, which is composed of hepatocytes with both

*For correspondence:
aliu@smu.edu.cn (AL);
mangmangli@smu.edu.cn (ML);
baixc15@smu.edu.cn (XB)

[†]These authors contributed equally to this work

Competing interest: The authors declare that no competing interests exist.

metabolic and regenerative capacities (*Fausto et al., 2006*). The unique regenerative capability of hepatocytes as differentiated cells after injury makes the liver an ideal in vivo model for studying the molecular mechanisms underlying quiescence exit. Recent studies have demonstrated that hepatocytes act like stem cells that possess regenerative potential to reenter the cell cycle and proliferate during liver regeneration (*Michalopoulos and Bhushan, 2021*; *Matsumoto et al., 2020*; *Chen et al., 2020*). Upon quiescence exit, a process also known as priming/initiation, hepatocytes favor proliferative capacity over metabolism to meet the rapid hepatic growth demand. However, how the active metabolic state is transcriptionally altered and modulated during the G0/G1 transition remains to be elucidated.

Methyl-CpG binding protein 2 (Mecp2), as a chromatin-binding protein (*Lee et al., 2020*; *Tillotson and Bird, 2020*), plays multiple roles in gene expression regulation, including transcriptional activation and repression, RNA splicing, chromatin remodeling, and regulation of chromatin architecture (*Ezeonwuka and Rastegar, 2014*). Given the high expression and pivotal role of Mecp2 in the brain, understanding the mechanisms of Mecp2 in neurological disorders such as Rett syndrome and autism has attracted intense interest (*Tillotson and Bird, 2020*; *Ip et al., 2018*; *Guy et al., 2011*). Furthermore, Mecp2 has also been identified as an oncogene highly expressed in several cancer types. Several lines of evidence suggest that the role of Mecp2 in malignancy mainly involves facilitation of cancer cell proliferation and inhibition of apoptosis (*Babbio et al., 2012*; *Neupane et al., 2016*; *Zhao et al., 2017*). The role of Mecp2 in quiescence exit and tissue regeneration and the underlying mechanisms have not been reported to the best of our knowledge.

In this study, using a mouse model of injury-induced liver regeneration and cellular models of quiescence exit, we intriguingly found that Mecp2 is a cell cycle-dependent protein that is drastically abated during the G0/G1 transition and gradually restored at further stages of cell cycle progression. A sharp decline in Mecp2 expression is essential for efficient quiescence exit in response to extrinsic stimuli using both in vivo and in vitro models. Additionally, the E3 ligase Nedd4 contributes to the ubiquitination and degradation of Mecp2 at quiescence exit, which modulates the pace of quiescence exit. Mechanistically, Mecp2 governs quiescence exit by transcriptionally orchestrating proliferative and metabolic gene expression, among which many nuclear receptor genes (NRs) emerge as novel Mecp2-activated genes in quiescent cells. Together, our findings identify a critical negative regulatory role for Mecp2 in quiescence exit and tissue regeneration, partially through targeting several NRs.

## Results

### Mecp2 is dynamically expressed during injury-induced liver regeneration

To screen for key regulators governing the initiation/priming phase of liver regeneration, we used the 2/3 partial hepatectomy (PHx) mouse model, a widely used in vivo model to study quiescence exit (*Mitchell and Willenbring, 2008*). Surprisingly, Mecp2, a well-known essential regulator of brain development, emerged as a dramatically repressed protein at the priming/initiation stage of PHx-induced liver regeneration. The expression kinetics of Mecp2 were examined at both the mRNA and protein levels in liver tissues at six time points after PHx, which cover the three phases of liver regeneration, namely priming/initiation, progression, and termination (*Fausto et al., 2006*; *Figure 1—figure supplement 1A*). The results showed that Mecp2 was remarkably reduced as early as 6 hr after PHx, was further decreased at 12 hr and 24 hr, but was restored at the 48 and 120 h time points (*Figure 1A and B*, *Figure 1—figure supplement 1B*). The decrease in the active histone mark H3K27ac (*Creyghton et al., 2010*) at the Mecp2 promoter was consistent with the reduced transcriptional activity within 36 hr post-PHx (*Figure 1—figure supplement 1C*). Notably, Mecp2 protein levels were decreased more dramatically than mRNA, suggesting a post-translational regulation of Mecp2 at the very early stage of liver regeneration. The early acute reduction of Mecp2 in hepatocytes during liver regeneration, mainly in nuclei, was further validated by immunofluorescence (IF) staining (*Figure 1C*) and immunohistochemistry (IHC) (*Figure 1D*). To confirm the quiescence exit-specific reduction of Mecp2 in hepatocytes, we assessed proliferation-associated proteins that are widely used to distinguish the G0 from the G1 phase, including phosphorylated retinoblastoma protein (pRb), Cyclin D1, and Ki67 (*Figure 1B and D*). The phosphorylation states of Rb, including unphosphorylated, hypo-phosphorylated, and hyper-phosphorylated Rb, can reflect G0, early G1,

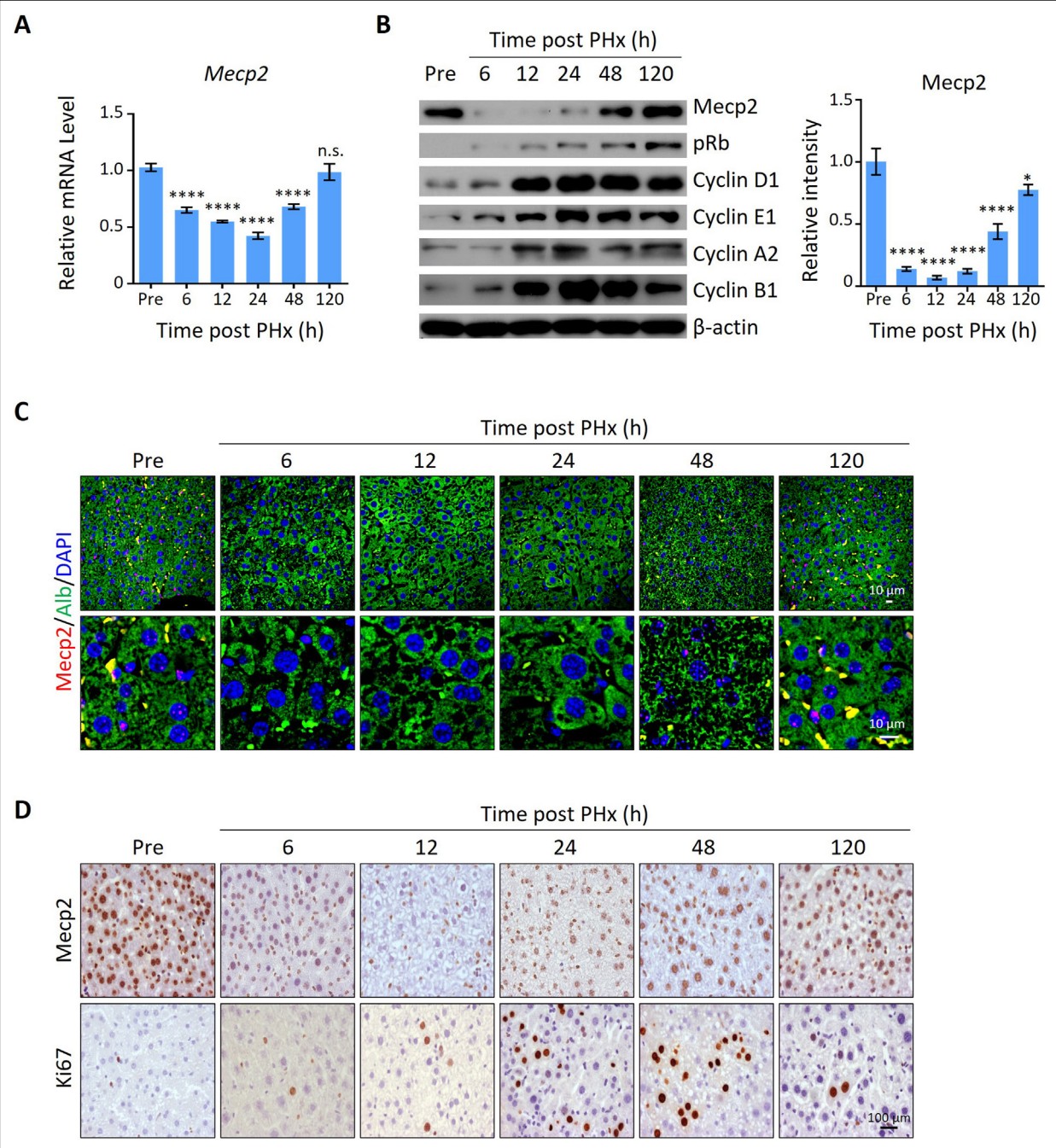

**Figure 1.** Mecp2 is immediately decreased in the liver after partial hepatectomy (PHx). (**A**) Real-time PCR to evaluate mRNA levels of Mecp2 at different time points after PHx. Data are presented as means ± SEM; n = 6. n.s., not significant; ****p<0.0001 by one-way ANOVA. (**B**) Western blotting (WB) showing the time course of protein levels of Mecp2, pRb, Cyclin D1, Cyclin E1, Cyclin A1, Cyclin B1, and β-actin in mouse livers after PHx. Right panel: quantification of Mecp2. Data are presented as means ± SEM; n = 3. n.s., not significant; ****p<0.0001; *p<0.05 by one-way ANOVA. (**C**) Representative immunofluorescence (IF) staining of Mecp2 (red) and Alb (green), together with DAPI (blue) for nuclei in liver sections at different time points after PHx. Lower panels: higher-magnification images. (**D**) Representative immunohistochemistry (IHC) images of liver tissues stained for Mecp2 or Ki67 at the indicated time points after PHx.

The online version of this article includes the following source data and figure supplement(s) for figure 1:

**Source data 1.** Mecp2 is immediately decreased in the liver after partial hepatectomy (PHx).

**Figure supplement 1.** Mecp2 is dynamically expressed during partial hepatectomy (PHx)-induced liver regeneration.

**Figure supplement 1—source data 1.** Mecp2 is dynamically expressed during partial hepatectomy (PHx)-induced liver regeneration.

and late G1 phases, respectively (*Narasimha et al., 2014*). The results showed that pRb was undetectable by 6 hr after PHx. Accordingly, Cyclin D1, which mediates the phosphorylation of Rb, was also maintained at extremely low levels during the very early stage of liver regeneration. Several other cyclins important for the G1/S stage (Cyclin E1) and the G2/M stage (Cyclin A2 and B1) were also expressed at low levels, further supporting the negative correlation between the amount of Mecp2 and cell cycle reentry (*Coller, 2007*; *Moser et al., 2018*; *Figure 1B*). Moreover, Ki67, a well-known proliferation marker, which is degraded and barely detected in the G0 phase but accumulates during cell proliferation (*Kim and Sederstrom, 2015*; *Miller et al., 2018*), showed an inverse correlation with Mecp2 in hepatocytes within 48 hr after PHx (*Figure 1D*). It is worth noting that the acute reduction in Mecp2 during quiescence exit was gradually restored by 48–120 hr post-PHx, suggesting the functional involvement of Mecp2 in active cell cycle phases. Together, these findings demonstrate the identification of Mecp2 as a new cell cycle-associated protein, which is highly expressed in quiescent hepatocytes, sharply decreased at the G0/G1 transition, and gradually restored at later stages of cell cycle progression during injury-induced liver regeneration.

## Mecp2 negatively regulates quiescence exit during PHx-induced liver regeneration

Given the acute reduction of Mecp2 during hepatocyte quiescence exit, we asked whether Mecp2 prevents the G0/G1 transition. To this aim, we generated hepatocyte-specific Mecp2 conditional knockout mice (Mecp2-cKO) by crossing control mice containing *Loxp* sites flanking exons 2 and 3 of the *Mecp2* gene (*Mecp2*^fl/fl^) with albumin (Alb)-Cre mice expressing Cre recombinase under the Alb promoter (*Figure 2—figure supplement 1A*). The successful Mecp2 depletion in the liver was confirmed by genotyping (*Figure 2—figure supplement 1B*) and measuring the amount of Mecp2 at both mRNA and protein levels in *Mecp2*^fl/fl^ and Mecp2-cKO mice (*Figure 2—figure supplement 1C and D*). Mecp2-cKO mice were viable and showed no obvious abnormalities in the liver compared to control littermates (*Figure 2—figure supplement 1E–G*). PHx triggered the decay of Mecp2 in livers at 6 and 48 hr after PHx, which reflected the G0/G1 transition and M phase, respectively, in the remaining hepatocytes undergoing the first round of the cell cycle after PHx (*Figure 2A–C*). The results of western blotting and corresponding quantification in Mecp2-cKO livers indicated that Mecp2 depletion promoted quiescence exit in hepatocytes after PHx (*Figure 2B*). IF staining for Ki67 showed that Mecp2-deficient livers contained more proliferating hepatocytes than controls, further supporting the enhanced G0/G1 transition in Mecp2-cKO mice (*Figure 2C*). Accordingly, liver regeneration was significantly enhanced at 6, 12, 24, and 48 hr after PHx based on the significantly higher liver index in the Mecp2-cKO mice than in the controls (*Figure 2D*). Therefore, hepatocyte-specific Mecp2 depletion accelerates quiescence exit in injury-induced regenerating livers.

We next tested whether overexpression (OE) of Mecp2 in hepatocytes has an adverse effect on cell cycle reentry after PHx. Intravenous injection of adeno-associated virus (AAV)-TBG-Mecp2 (*Greig et al., 2018*) reinforced the levels of Mecp2 in hepatocytes compared to empty vector (EV) control cells (*Figure 2E and F*). Decreased protein levels of pRb and Cyclin D1, as well as lower levels of Ki67 in nuclei from Mecp2-overexpressing hepatocytes, were observed within 48 hr after PHx (*Figure 2F and G*). Expectedly, Mecp2 OE significantly delayed cell cycle reentry and resulted in a decreased liver index within 24 hr after PHx (*Figure 2H*). Therefore, Mecp2 OE negatively regulates quiescence exit in hepatocytes after PHx.

To further confirm the specificity of Mecp2 on the regulation of quiescence exit, we performed rescue experiments using AAV-mediated Mecp2 OE in Mecp2-cKO livers (*Figure 2I and J*). The restoration of Mecp2 in Mecp2-depleted hepatocytes was accompanied by increased expression of cell cycle regulators and earlier appearance of Ki67 compared to the EV controls (*Figure 2J and K*). In addition, the increased liver index caused by the loss of Mecp2 was significantly compromised by Mecp2 restoration (*Figure 2L*). Therefore, forced restoration of Mecp2 rescues Mecp2 loss-induced accelerated quiescence exit.

Notably, the modest but significant changes in liver regeneration caused by the manipulation of Mecp2 in the first 2 d after PHx disappeared 5 d post-PHx (*Figure 2—figure supplement 1H–J*), indicating the involvement of Mecp2 in not only quiescence exit but also in later stages of cell proliferation. Taken together, our in vivo analyses demonstrate that the rapid reduction of Mecp2 in hepatocytes is essential for efficient quiescence exit during injury-induced liver regeneration.

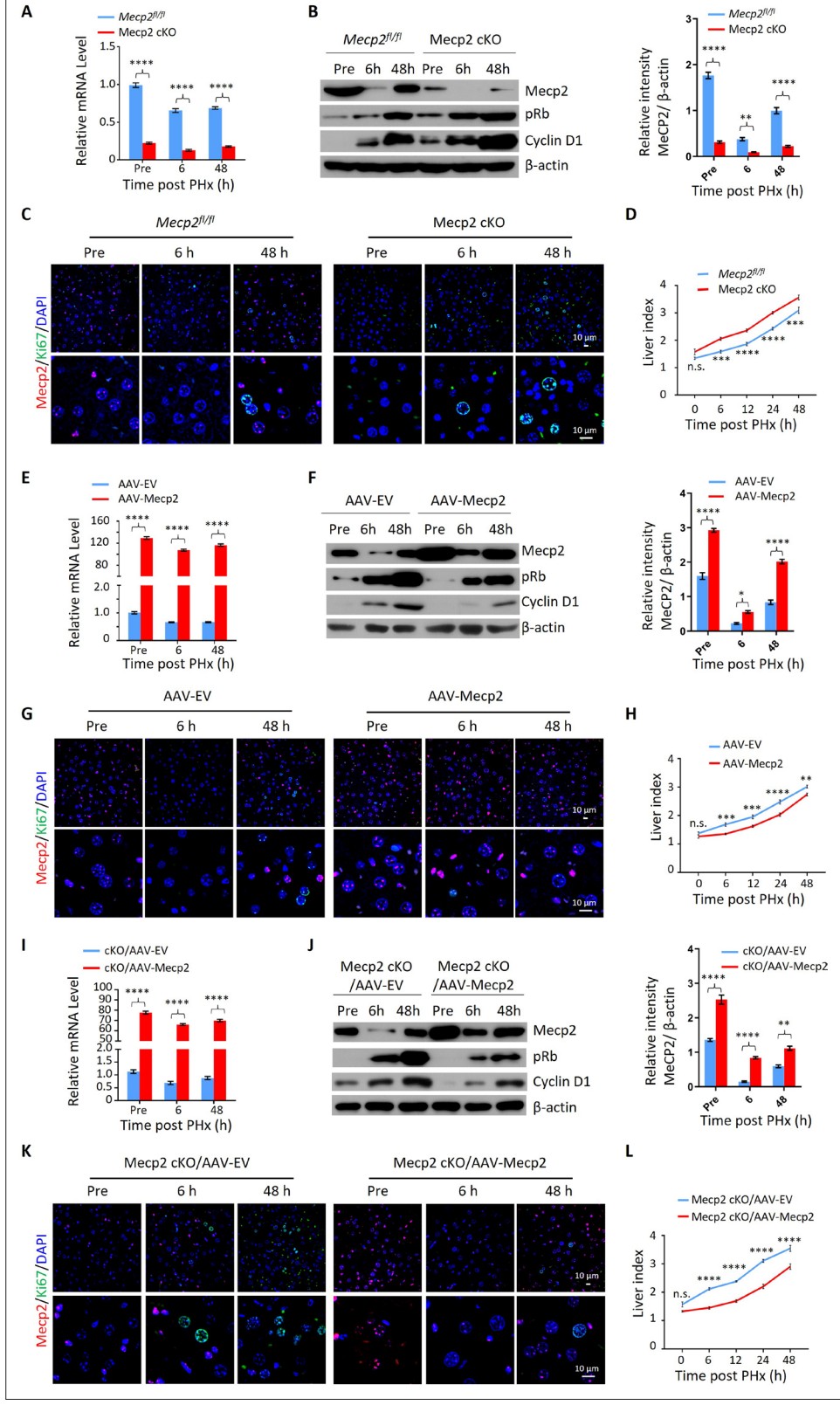

**Figure 2.** Mecp2 fine-tunes quiescence exit in hepatocytes after partial hepatectomy (PHx) in vivo. (**A–D**) Liver regeneration in *Mecp2^{fl/fl}* and Mecp2 cKO mice after PHx. (**A**) Real-time PCR to measure mRNA levels of *Mecp2*. The effects and corresponding quantification of Mecp2 KO on quiescence exit and liver regeneration were assessed by western blotting (WB) of Mecp2, pRb, and Cyclin D1 (**B**), immunofluorescence (IF) staining of Mecp2 (red) and Ki67

*Figure 2 continued on next page*

*Figure 2 continued*

(green) in liver sections (**C**), and liver index of control and Mecp2 cKO mice (**D**) at the indicated time points. (**E–H**) Liver regeneration in *Mecp2*[fl/fl] livers without (AAV-EV) or with AAV-mediated Mecp2 OE (AAV-Mecp2) after PHx. AAV, adeno-associated virus; EV, empty vector. (**E**) Real-time PCR to measure mRNA levels of *Mecp2*. (**F–H**) The effects and corresponding quantification of Mecp2 OE on quiescence exit and liver regeneration were assessed by WB of Mecp2, pRb, and Cyclin D1 (**F**), IF staining of Mecp2 (red) and Ki67 (green) in liver sections (**G**), and liver index at the indicated time points (**H**). (**I–L**) Liver regeneration in Mecp2 cKO livers without (Mecp2 cKO/AAV-EV) or with AAV-mediated Mecp2 restoration (Mecp2 cKO/AAV-Mecp2) after PHx. (**I**) Real-time PCR to measure mRNA levels of *Mecp2*. (**J–L**) The effects and corresponding quantification of Mecp2 restoration on quiescence exit and liver regeneration in Mecp2 cKO livers were assessed by WB of Mecp2, pRb and Cyclin D1 (**J**), IF staining of Mecp2 (red) and Ki67 (green) in liver sections (**K**), and liver index (**L**) at the indicated time points. Data are presented as means ± SEM. In (**A, E, I**), n = 6; (**B, F, J**), n = 3; in (**D, H, L**), n = 5 mice/group. n.s., not significant; *p<0.05; **p<0.01; ***p<0.001; ****p<0.0001 by two-way ANOVA.

The online version of this article includes the following source data and figure supplement(s) for figure 2:

**Source data 1.** Mecp2 fine-tunes quiescence exit in hepatocytes after partial hepatectomy (PHx) in vivo.

**Figure supplement 1.** Modulation of Mecp2 expression in the mouse liver.

**Figure supplement 1—source data 1.** Modulation of Mecp2 expression in the mouse liver.

## Acute reduction of Mecp2 is universal for quiescence exit in cellular models

We then asked whether the expression pattern of Mecp2 during the quiescence-proliferation transition is universal in cells. To this end, we released three types of cultured cells, including 3T3 mouse embryonic fibroblasts, mouse hippocampal neuronal HT22 cells, and human primary umbilical vein endothelial cells (HUVECs), from quiescence induced by two classical signals, that is, serum starvation (SS) and contact inhibition (CI) *Coller et al., 2006*. The cell cycle analysis in 3T3 cells showed that more than 90% of cells resided in the G0/G1 phase in response to SS (*Figure 3A and B*) or CI (*Figure 3F and G*), indicating the successful induction of quiescence. The expression kinetics revealed that serum restimulation (SR)- and/or CI loss (CIL)-induced cell cycle reentry of quiescent 3T3 cells resulted in a dramatic decrease in Mecp2 at the G0/G1 transition and its gradual restoration during cell cycle progression, resembling that seen after PHx-induced initiation of hepatocellular regeneration (*Figure 3C, D, H, and I*). The time course of IF staining for Ki67 and Mecp2 showed extremely low levels of Ki67 signals in nuclei within 6 hr after SR (*Figure 3E*) or 24 hr after CIL (*Figure 3J*), further supporting the reduction in Mecp2 at quiescence exit in 3T3 cells. Similar results were obtained in both HT22 cells (*Figure 3—figure supplement 1A–H*) and HUVECs (*Figure 3—figure supplement 1I–P*). Taken together, this evidence suggests that the acute reduction in Mecp2 is a general phenomenon at the G0/G1 transition.

## Acute reduction of Mecp2 is essential for efficient quiescence exit in cells

To determine whether the functional relationship between acute Mecp2 reduction and quiescence exit also exists in cells other than hepatocytes, we first assessed the effect of siRNA-mediated Mecp2 knockdown (KD) on the SR-induced quiescence exit of 3T3 cells (*Figure 4A*). Western blotting showed that Mecp2 depletion led to an earlier induction of pRb, Cyclin D1, as well as cyclin A, Cyclin B, and Cyclin E proteins (*Figure 4B*, *Figure 4—figure supplement 1A and C*). The accelerated cell cycle reentry in Mecp2 KD cells was also reflected by the earlier appearance of Ki67 (*Figure 4C*). Additionally, we used Ki67 and propidium iodide (PI) double staining followed by flow cytometry to quantify G0 cells (Ki67⁻ with 2N DNA content) at early time points after quiescence exit with or without Mecp2 depletion. The results showed that Mecp2 KD significantly reduced G0 cells compared to negative control siRNA (NC si)-treated cells after SR (*Figure 4D*). As such, about 70% of control cells and over 90% of Mecp2 KD cells reentered the cell cycle at 6 hr post-SR, indicating that Mecp2 KD accelerated quiescence exit in 3T3 cells. Thus, enhanced reduction of Mecp2 stimulates exit from quiescence.

We then asked whether increased Mecp2 expression could postpone quiescence exit. To test this, we forced Mecp2 OE in 3T3 cells through lentiviral transduction (*Figure 4E and F*). Compared to the EV, Mecp2 OE resulted in the delayed quiescence exit phenotype upon SR, as evidenced by

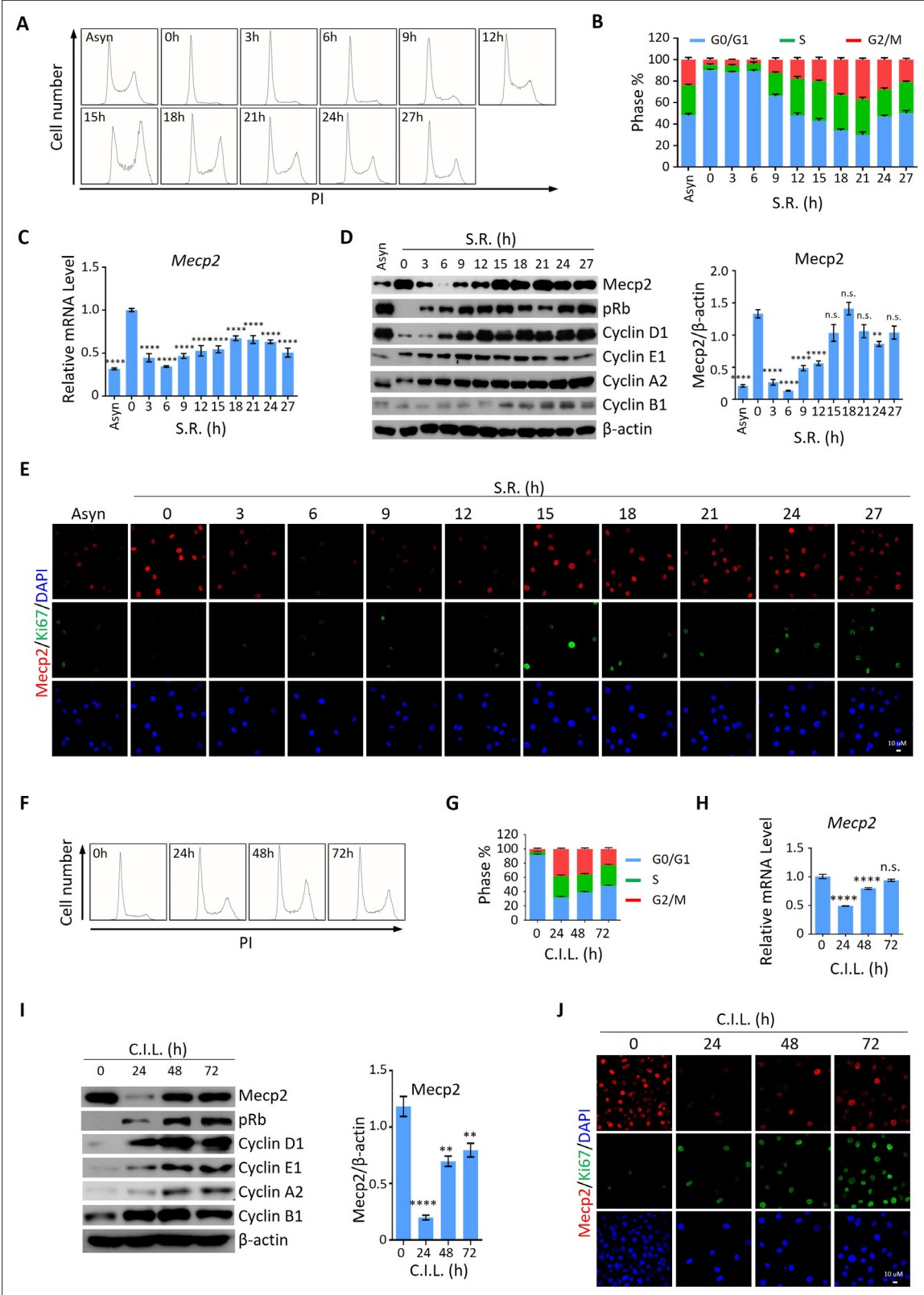

**Figure 3.** Mecp2 is immediately decreased during quiescence exit in cellular models. (**A, B**) Representative histograms of propidium iodide (PI) staining of either asynchronized (Asyn) proliferating or starvation-induced quiescent 3T3 cells after serum restimulation (SR) in (**A**) and statistical analysis of cell cycle distribution (**B**). (**C**) Real-time PCR to examine mRNA levels of Mecp2. Data are presented as means ± SEM; n = 9. n.s., not significant; ****p<0.0001 by one-way ANOVA. (**D**) Western blotting (WB) of Mecp2, pRb, Cyclin D1, Cyclin E1, Cyclin A1, and Cyclin B1 in quiescent 3T3 cells

*Figure 3 continued on next page*

*Figure 3 continued*

upon SR. Right panel: quantification of Mecp2. Data are presented as means ± SEM; n = 3. n.s., not significant; ****p<0.0001 by one-way ANOVA. (**E**) Representative immunofluorescence (IF) staining of Mecp2 (red) and Ki67 (green), together with DAPI (blue) in Asyn and quiescent 3T3 cells upon SR. (**F, G**) Representative histograms of PI staining of contact inhibition (CI)-induced quiescent 3T3 cells after CI loss (CIL) (**F**) and statistical analysis of cell cycle distribution (**G**) at the indicated time points. (**H**) Real-time PCR to examine mRNA levels of *Mecp2* in 3T3 cells released from CI-induced quiescence. Data are presented as means ± SEM; n = 6. n.s., not significant; ****p<0.0001 by one-way ANOVA. (**I**) WB of Mecp2, pRb, Cyclin D1, Cyclin E1, Cyclin A1, and Cyclin B1 in quiescent 3T3 cells upon CIL. Right panel: quantification of Mecp2. Data are presented as means ± SEM; n = 3. n.s., not significant; ****p<0.0001 by one-way ANOVA. (**J**) Representative IF staining of Mecp2 and Ki67 in 3T3 cells released from CI-induced quiescence.

The online version of this article includes the following source data and figure supplement(s) for figure 3:

**Source data 1.** Mecp2 is immediately decreased during quiescence exit in cellular models.

**Figure supplement 1.** Mecp2 is dynamically expressed during quiescence exit in both HT22 and human umbilical vein endothelial cells (HUVECs).

**Figure supplement 1—source data 1.** Mecp2 is dynamically expressed during quiescence exit in both HT22 and human umbilical vein endothelial cells (HUVECs).

decreased expression of pRb, Cyclin D1, and other cell cycle protein, such as Cyclin A2, Cyclin B1, and Cyclin E1 (*Figure 4F*, *Figure 4—figure supplement 1B and D*), delayed induction of Ki67 (*Figure 4G*), and a twofold increase in the proportion of cells residing in the G0 phase (*Figure 4H*). Notably, neither Mecp2 KD nor OE significantly affected the number of G0 cells without SR (*Figure 4D and H*), suggesting that Mecp2 functions upon receiving extracellular mobilization signals. Taken together, these results indicate that acute Mecp2 reduction at the G0/G1 transition is required for efficient quiescence exit.

## Nedd4 contributes to Mecp2 degradation and regulates quiescence exit

Given the more rapid decay of Mecp2 at the protein compared to the mRNA level during the quiescence-proliferation transition, we speculated that Mecp2 is targeted by post-translational regulation. This hypothesis was supported by proteasome inhibition with the proteasome inhibitor MG132, which attenuated the reduction of Mecp2 in quiescent cells after SR (*Figure 5—figure supplement 1A*). In contrast, the lysosome inhibitor chloroquine had no impact on the degradation of MeCP2 protein (*Figure 5—figure supplement 1B*). To identify the proteins that regulate Mecp2 degradation during the G0/G1 transition, we performed immunoprecipitation followed by mass spectrometry (IP-MS) using Mecp2 antibody in quiescent 3T3 cells treated with or without SR (*Figure 5—figure supplement 1C*). A total of 647 proteins were identified as putative Mecp2 interactors. We were particularly interested in the proteins involved in proteasome-mediated ubiquitin-dependent protein catabolic process, which was one of the enriched Gene Ontology (GO) items in the Mecp2 interactome (*Supplementary file 1*). Among the candidate genes, we identified the ubiquitin ligase 'neuronal precursor cell developmentally downregulated 4-1' (Nedd4) that specifically interacted with endogenous Mecp2 by reciprocal IP-WB in 3T3 cells (*Figure 5A*). Co-IP using Mecp2 antibody revealed that Mecp2-associated ubiquitin and Nedd4 were dramatically increased at 3 hr and 6 hr post-SR (*Figure 5B*), suggesting that Nedd4 interacts with Mecp2 to induce the polyubiquitination and degradation of Mecp2 upon cell cycle reentry.

Given the important role of Nedd4 in post-translational regulation of Mecp2, we reasoned that despite regulating internalization of major growth factor receptors involved in liver regeneration (*Bachofner et al., 2017*), Nedd4-mediated degradation of Mecp2 may be a new mechanism through which Nedd4 contributes to liver regeneration. To test this hypothesis, we assessed the effects of Nedd4 on SR-induced quiescence exit of 3T3 cells. Knocking down Nedd4 using siRNA replenished the remarkable reduction of Mecp2 at the early stages of quiescence exit, and thus resulted in lower levels of proliferation markers such as pRb, Cyclin D1, and Ki67 (*Figure 5C–E*). Nedd4 deficiency significantly retained cells in the G0 phase upon SR (*Figure 5F*). On the contrary, Nedd4 OE significantly enhanced the degradation of Mecp2 and accordingly accelerated the quiescence exit, which mimicked the effect of Mecp2 depletion (*Figure 5G–J*). Thus, Nedd4 interacts with Mepc2 and regulates quiescence exit partially through post-translational regulation of Mecp2 upon quiescence exit.

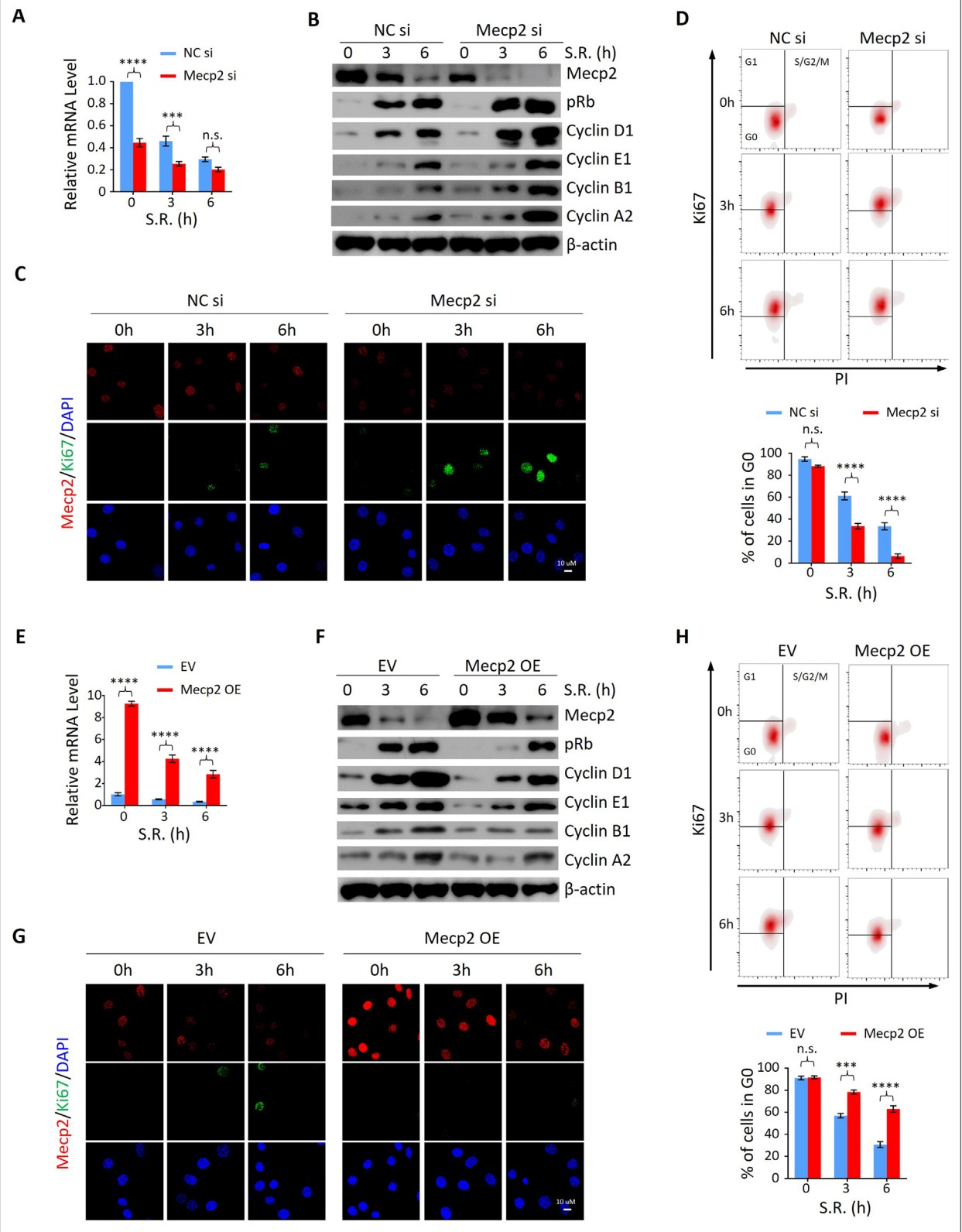

**Figure 4.** Mecp2 negatively regulates the G0/G1 transition in the cellular model of serum restimulation (SR)-induced quiescence exit. (**A**) Real-time PCR showing the mRNA levels of *Mecp2* in 3T3 cells transfected with negative control siRNA (NC si) or Mecp2 siRNA (Mecp2 si) at the early stages of SR-induced cell cycle reentry. Data are presented as means ± SEM; n = 6. n.s., not significant; ***p<0.001; ****p<0.0001 by two-way ANOVA. (**B**) Western blotting (WB) of Mecp2, pRb, Cyclin D1, Cyclin E1, Cyclin A2, and Cyclin B1 in control and Mecp2 knockdown (KD) 3T3 cells released from serum

*Figure 4 continued on next page*

*Figure 4 continued*

starvation (SS)-induced quiescence at the indicated time points. (**C**) Representative immunofluorescence (IF) staining of Mecp2 and Ki67 in control and Mecp2 KD 3T3 cells upon SR-induced quiescent exit. (**D**) Ki67 and propidium iodide (PI) double staining followed by flow cytometry showing cell cycle profiles of 3T3 cells transfected with NC or Mecp2 siRNA upon SR-induced quiescence exit. Cells in G0, G1, and S/G2/M phases were defined by Ki67−/2N DNA content, Ki67+/2N DNA content and >2N DNA content population, respectively. Lower panel: quantification of the percentage of 3T3 cells in the G0 phase. Data are presented as means ± SEM; n = 3. n.s., not significant; ****p<0.0001 by two-way ANOVA. (**E**) Real-time PCR showing the mRNA levels of *Mecp2* in 3T3 cells transduced with the empty vector (EV) or the vector overexpressing Mecp2 (Mecp2 overexpression [OE]) at the early stages of quiescence exit. Data are presented as means ± SEM; n = 3. ****p<0.0001 by two-way ANOVA. (**F**) WB of Mecp2, pRb, Cyclin D1, Cyclin E1, Cyclin A2, and Cyclin B1 in control and Mecp2 OE 3T3 cells released from SS-induced quiescence at the indicated time points. (**G**) Representative IF staining of Mecp2 and Ki67 in quiescent control and Mecp2 OE 3T3 cells upon SR. (**H**) Representative flow cytometry plots of Ki67/PI double staining in control and Mecp2 OE 3T3 cells upon SR-induced quiescence exit. Lower panel: quantification of proportion of 3T3 cells in the G0 phase. Data are presented as means ± SEM; n = 3. n.s., not significant; ***p<0.001; ****p<0.0001 by two-way ANOVA.

The online version of this article includes the following source data and figure supplement(s) for figure 4:

**Source data 1.** Mecp2 negatively regulates the G0/G1 transition in the cellular model of serum restimulation (SR)-induced quiescence exit.

**Figure supplement 1.** Mecp2 negatively regulates the G0/G1 transition in the cellular model of serum restimulation (SR)-induced quiescence exit.

**Figure supplement 1—source data 1.** Mecp2 negatively regulates the G0/G1 transition in the cellular model of serum restimulation (SR)-induced quiescence exit.

## Mecp2 slows quiescence exit by transcriptionally activating metabolism-associated genes while repressing proliferation-associated genes

It has been well established that Mecp2 transcriptionally regulates gene expression by binding methylated CpG islands and chromatin proteins in the brain (*Lee et al., 2020*; *Nan et al., 1997*; *Rube et al., 2016*). However, little is known about the transcriptional targets of Mecp2 during hepatocyte quiescence exit in the regenerating liver. To decipher the molecular mechanisms underlying Mecp2-regulated quiescence exit, we performed RNA-seq combined with chromatin immunoprecipitation followed by next-generation sequencing (ChIP-seq) to identify the Mecp2-dependent transcriptome genome-wide during the very early stage of liver regeneration (*Figure 6—figure supplement 1C*). RNA-seq and comparative analyses in control *Mecp2^fl/fl* and Mecp2-cKO mice livers before and 6 hr after PHx revealed 3048 Mecp2-dependent genes that were differentially expressed in a Mecp2-dependent manner. Meanwhile, we mapped the binding landscape of Mecp2 in *Mecp2^fl/fl* livers before and after PHx using ChIP-seq by filtering out peaks identified in Mecp2-cKO livers (*Figure 6—figure supplement 1C*). It has been reported that Mecp2 occupies a large proportion of the genome in the brain due to its methyl-CpG (mCpG)-binding preference (*Lee et al., 2020*; *Rube et al., 2016*; *Lagger et al., 2017*). Similarly, we identified a total of 14,640 and 15,350 Mecp2-binding genes before and after PHx in the Mecp2 control liver, respectively (*Figure 6—figure supplement 1A and B*). To further identify putative Mecp2-direct target genes, we integrated Mecp2-dependent genes with Mecp2-binding genes (*Figure 6—figure supplement 1C*). As a result, there were 2658 Mecp2 direct target genes, in which 537 were PHx-activated and 2121 were PHx-repressed genes (*Figure 6A*). GO analysis showed that PHx-activated Mecp2 targets, which are either silent or basally expressed in quiescent hepatocytes, were highly enriched in proliferation-associated biological processes such as ribosome biogenesis, rRNA metabolic process, ncRNA metabolic process, and regulation of transcription by RNA polymerase I, whereas PHx-repressed Mecp2 targets, which are highly expressed in quiescent hepatocytes, were associated with several metabolic processes including carboxylic acid catabolic process, cellular amino acid metabolic process, fatty acid metabolic process, and steroid metabolic process (*Figure 6B*). Notably, among PHx-repressed genes, several NRs were newly identified as Mecp2 direct targets, such as *Nr2f6*, *Nr3c1*, *Nr1h3*, *Nr1i3*, *Nr6a1*, *Rxrg*, *Rara*, *Nr4a1*, *Srebf1*, and *Ppard* (*Figure 6A*). To gain insights into the relevance between Mecp2 occupancy and the differential expression of Mecp2 direct targets, we interrogated the binding strength of Mecp2 in the promoter-proximal regions (within 3 kb of the transcription start sites [TSS]) and defined gene regions (±25 kb around the TSS and transcription end sites [TES]) (*Figure 6—figure supplement 1A*). Intriguingly, Mecp2 occupancy at these regions was not apparently altered after PHx, which was inconsistent with its protein levels, suggesting that the majority of Mecp2 is not tightly associated with the genome in

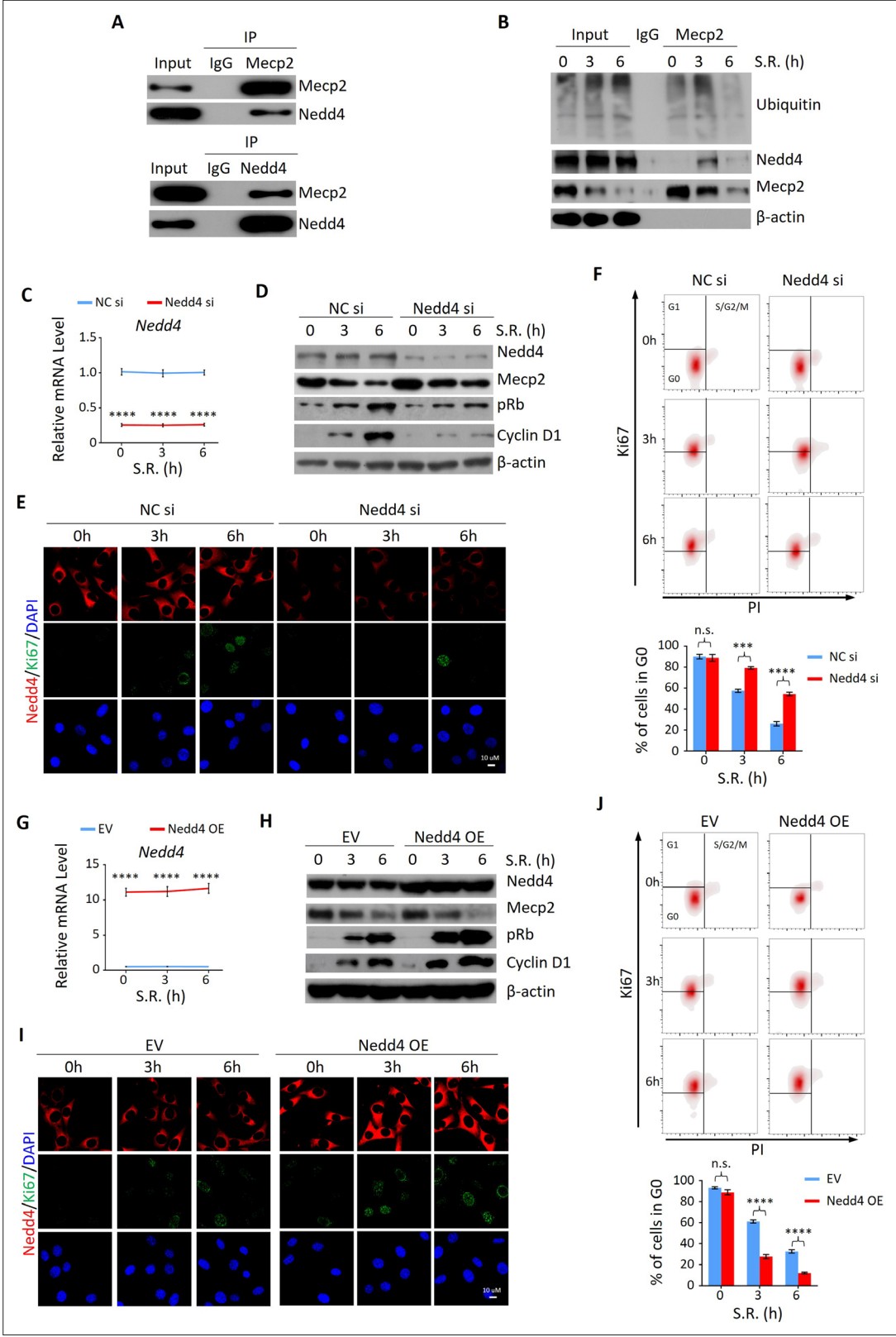

**Figure 5.** Nedd4 interacts with Mecp2 and affects quiescence exit by facilitating Mecp2 degradation. (**A**) Reciprocal immunoprecipitation-western blotting (IP-WB) analysis to validate the interaction between endogenous Mecp2 and Nedd4. (**B**) Co-IP of Mecp2, ubiquitin, and Nedd4 in quiescent 3T3 cells during serum restimulation (SR)-induced quiescence exit. (**C**) Real-time PCR showing siRNA-mediated Nedd4 knockdown (KD) in

*Figure 5 continued on next page*

*Figure 5 continued*

3T3 cells upon SR-induced quiescence exit. Data are presented as means ± SEM; n = 5. ****p<0.0001 by two-way ANOVA. (**D–F**) The effect of Nedd4 KD on quiescent exit in 3T3 cells determined by WB (**D**), immunofluorescence (IF) staining of Ki67 and Nedd4 (**E**), and Ki67/PI staining followed by flow cytometry (**F**) at the indicated time points. Lower panel in (**F**): quantification of the percentage of 3T3 cells in the G0 phase. Data are presented as means ± SEM; n = 3. n.s., not significant; ***p<0.001, ****p<0.0001 by two-way ANOVA. (**G**) Real-time PCR showing Nedd4 overexpression (OE) in 3T3 cells upon SR-induced quiescence exit. Data are presented as means ± SEM; n = 5. ****p<0.0001 by two-way ANOVA. (**H–J**) The effect of Nedd4 OE on quiescent exit in 3T3 cells determined by WB (**H**), IF staining of Ki67 and Nedd4 (**I**), and Ki67/PI staining (**J**) at the indicated time points. Data are presented as means ± SEM; n = 3. n.s., not significant; ****p<0.0001 by two-way ANOVA.

The online version of this article includes the following source data and figure supplement(s) for figure 5:

**Source data 1.** Nedd4 interacts with Mecp2 and affects quiescence exit by facilitating Mecp2 degradation.

**Figure supplement 1.** Nedd4 interacts with Mecp2.

**Figure supplement 1—source data 1.** Nedd4 interacts with Mecp2.

quiescent hepatocytes and Mecp2 may recruit other factor(s) to achieve the differential transcriptional outcomes in response to extrinsic stimuli (*Figure 6C*).

Because the liver is a key metabolic organ, we paid particular attention to Mecp2-regulated NRs, whose alteration may result in a shift in the balance between metabolism and proliferation. The Mecp2-dependent transcriptional repression of 10 NRs upon hepatic resection was validated using real-time qPCR in both Mecp2 control and cKO livers (*Figure 6D*). Notably, the mRNA levels of NRs were significantly higher in *Mecp2^{fl/fl}* than in Mecp2-cKO livers before PHx, suggesting that Mecp2 contributes to the transcriptional activation of NRs in normal livers while Mecp2 degradation upon PHx leads to their deactivation. We also confirmed the repression of NRs in SR-induced 3T3 cell quiescence exit (*Figure 6—figure supplement 1D*). We did not detect the expression of *Nr1i3* and *Rxrg* in quiescent 3T3 cells either before or after G0 exit, and failed to validate the repression of *Nr4a1*, suggesting that Mecp2-mediated transcriptional repression of NRs may vary with cell type and/or mobilization signals. Thus, these results suggest that Mecp2 plays a negative regulatory role during quiescence exit by activating metabolism-associated genes while repressing proliferation-associated genes in quiescent cells.

## Abolishing Mecp2-activated NRs promotes G0/G1 transition in vitro and in vivo

To interrogate the functional relevance of Mecp2-mediated repression of NRs to quiescence exit, we selected two candidate genes for further investigation, *Rara* and *Nr1h3*. Retinoic acid (RA) has long been recognized as a liver mitogen required for normal liver regeneration (*Bushue and Wan, 2009*). Rara and Rarb as receptors for RA have been recently identified to mediate RA-induced hepatocyte proliferation after PHx (*Liu et al., 2014*). Cholesterol is another important regulator of cell proliferation. Nr1h3 (LXR) is highly expressed in the liver and has recently been reported to reduce hepatocyte proliferative capacity during PHx-induced regeneration by regulating genes involved in lipid and cholesterol homeostasis (*Lo Sasso et al., 2010*; *Willy et al., 1995*). We first performed ChIP-qPCR to confirm the significantly decreased binding intensity of Mecp2 at proximal promoter regions of both *Rara* and *Nr1h3* upon exit quiescence (*Figure 7A*). Using lentivirus-mediated gene knockdown, we then tested whether individual disruption of two candidate NRs might affect SR-induced quiescence exit in 3T3 cells (*Figure 7B*). The results of western blotting showed that depletion of either *Nr1h3* or *Rara* significantly accelerated the G0/G1 transition, as measured by the expression of pRb and Cyclin D1 (*Figure 7C*) and by flow cytometry (*Figure 7D*), mimicking the Mecp2 KD phenotype. Therefore, Mecp2 prevents quiescence exit, at least in part, by repressing Rara and Nr1h3.

We then asked whether depletion of Rara or Nr1h3 can further promote quiescence exit in Mecp2-cKO livers. We performed AAV-mediated gene knockdown to target *Rara* and *Nr1h3* in Mecp2-cKO livers using short hairpin RNAs (shRNAs) (*Figure 7E*). Because of the limited number of cKO animals, we could only assess the effects at 6 hr post-PHx. The results showed that knockdown of either *Rara* or *Nr1h3* in combination with Mecp2-cKO can modestly but significantly further accelerate quiescence exit during PHx-induced liver regeneration (*Figure 7F–H*). Therefore, for the first time to the best of our knowledge, this study has revealed a positive correlation between the repression of

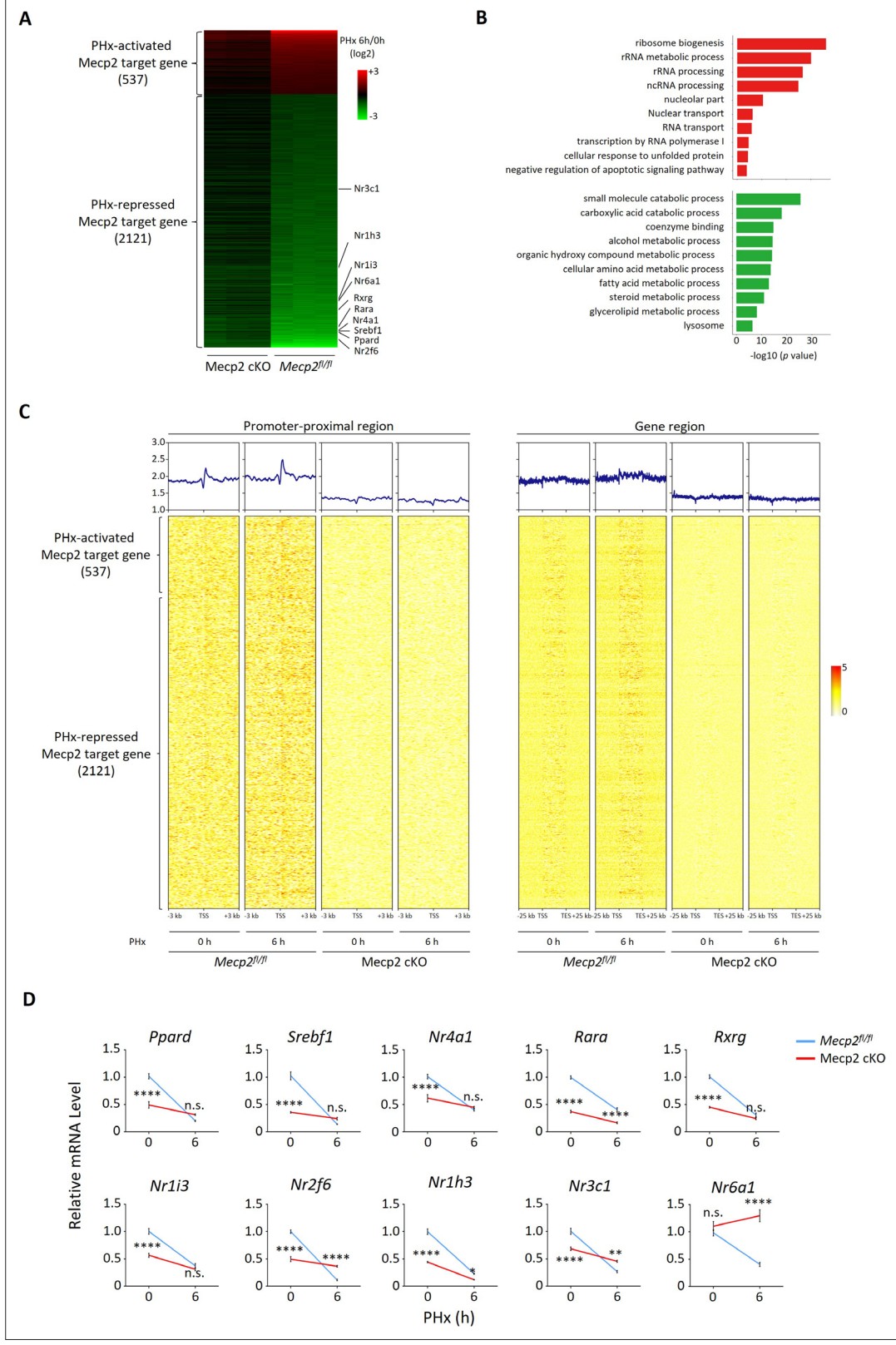

**Figure 6.** Mecp2 transcriptionally regulates quiescence exit. (**A**) Heatmap of Mecp2 direct target genes at the early stage of liver regeneration rank-ordered by their gene expression fold change. (**B**) The top 10 most significantly overrepresented Gene Ontology (GO) terms for the partial hepatectomy (PHx)-activated (red) and PHx-repressed (green) Mecp2 target genes. (**C**) Heatmaps depicting ChIP-seq enrichment of Mecp2 at the promoter-proximal

*Figure 6 continued on next page*

*Figure 6 continued*

region (3 kb away from transcription start sites [TSS]) and the defined gene region of Mecp2 target genes in *Mecp^fl/fl^* and Mecp2 cKO livers before and 6 hr post-PHx. Genes are rank-ordered according to the fold change of expression. (**D**) Real-time PCR validation of PHx-repressed NRs in *Mecp2^fl/fl^* and Mecp2 cKO livers upon PHx. Data are presented as means ± SEM; n = 5. n.s., not significant; **p<0.01; ****p<0.0001 by two-way ANOVA.

The online version of this article includes the following source data and figure supplement(s) for figure 6:

**Source data 1.** Mecp2 transcriptionally regulates quiescence exit.

**Figure supplement 1.** Related to *Figure 6*.

**Figure supplement 1—source data 1.** Raw data related to *Figure 6—figure supplement 1*.

Mecp2-activated NRs and quiescence exit, and has identified novel roles of Rara and Nr1h3 in regulating quiescence exit in vitro and in vivo.

## Discussion

The accurate transition from quiescence to the active cell cycle is crucial for the control of eukaryotic cell proliferation and adult stem cell-mediated tissue homeostasis and regeneration after injury. Conversely, dysregulation of quiescence exit may compromise tissue integrity and lead to oncogenesis. In this work, we sought to explore the general molecular mechanisms that regulate quiescence by focusing on Mecp2, a multifunctional protein with a broad spectrum of activities. Using genetic mouse models, cellular models, and genome-wide approaches, we uncovered a regulatory capacity of Mecp2 in quiescence exit (*Figure 7I*). In quiescent cells, Mecp2 is maintained at relative high levels and serves as both a transcriptional activator and repressor. It binds to and activates metabolic genes, such as several NRs, while repressing proliferation-associated genes in quiescent and metabolically hyperactive hepatocytes. In response to extrinsic stimuli, such as injury and mitogenic stimulation, the protein levels of Mecp2 are acutely decreased by both transcriptional repression and Nedd4-mediated ubiquitination. The remarkable reduction of Mecp2 releases the transcriptional repression of proliferation-associated genes while compromising the activation of metabolic genes in order to satisfy a rapidly regenerating demand of the remaining hepatocytes, eventually leading to quiescence exit and cell cycle progression. The transient repression of Mecp2 during quiescence exit and its restoration during further cell cycle progression probably serve as means to avoid overinhibition of metabolism and guarantee the appropriate metabolic adaption required for cell proliferation. Therefore, our results suggest that quiescent cells employ Mecp2 to balance the needs of cell division and metabolism upon receiving extrinsic signals.

Although it has been reported that Mecp2 null mice develop fatty liver (*Kyle et al., 2016*), the Mecp2-cKO mice used in our study did not demonstrate obvious abnormalities, such as necrosis or liver damage when we performed PHx (*Figure 2—figure supplement 1G*). Based on our comparative analysis of RNA-seq data from control and Mecp2 cKO livers before PHx, only 90 upregulated and 128 downregulated genes (log2 |FC|>1.5, p<0.05) were identified, which did not enrich any GO terms with adjusted p<1 × 10⁻³, further supporting the notion that liver-specific deletion of Mecp2 does not cause liver abnormalities in 3-month-old mice. This allowed us to study liver regeneration in mice with non-damaged livers and avoid any defects prior to injury. It is worth noting that, other than accelerated cell cycle reentry, Mecp2 cKO hepatocytes also displayed a modest increase in cell size with enlarged nuclei relative to control cells (*Figure 2—figure supplement 1G*), implying the attenuation of mitosis.

It has been well documented that both overexpression and depletion of Mecp2 have deleterious effects on neuronal homeostasis (*Na et al., 2013*), and thus, tight regulation of Mecp2 protein levels is critical for its physiological functions. In addition, several studies have demonstrated that many Rett syndrome-causing mutations in the methyl-CpG binding domain not only compromise DNA binding capacity of Mecp2 but also reduce its protein stability, implying the relevance of protein stability in Mecp2 dysfunction (*Tillotson and Bird, 2020*; *Chen et al., 2017*; *Brown et al., 2016*). Mecp2 has been shown to undergo various post-translational modifications, including phosphorylation, acetylation, ubiquitination, and sumoylation, which may also affect protein stability (*Ausió et al., 2014*). Protein degradation via ubiquitination is the most prevalent recycling machinery used by cells.

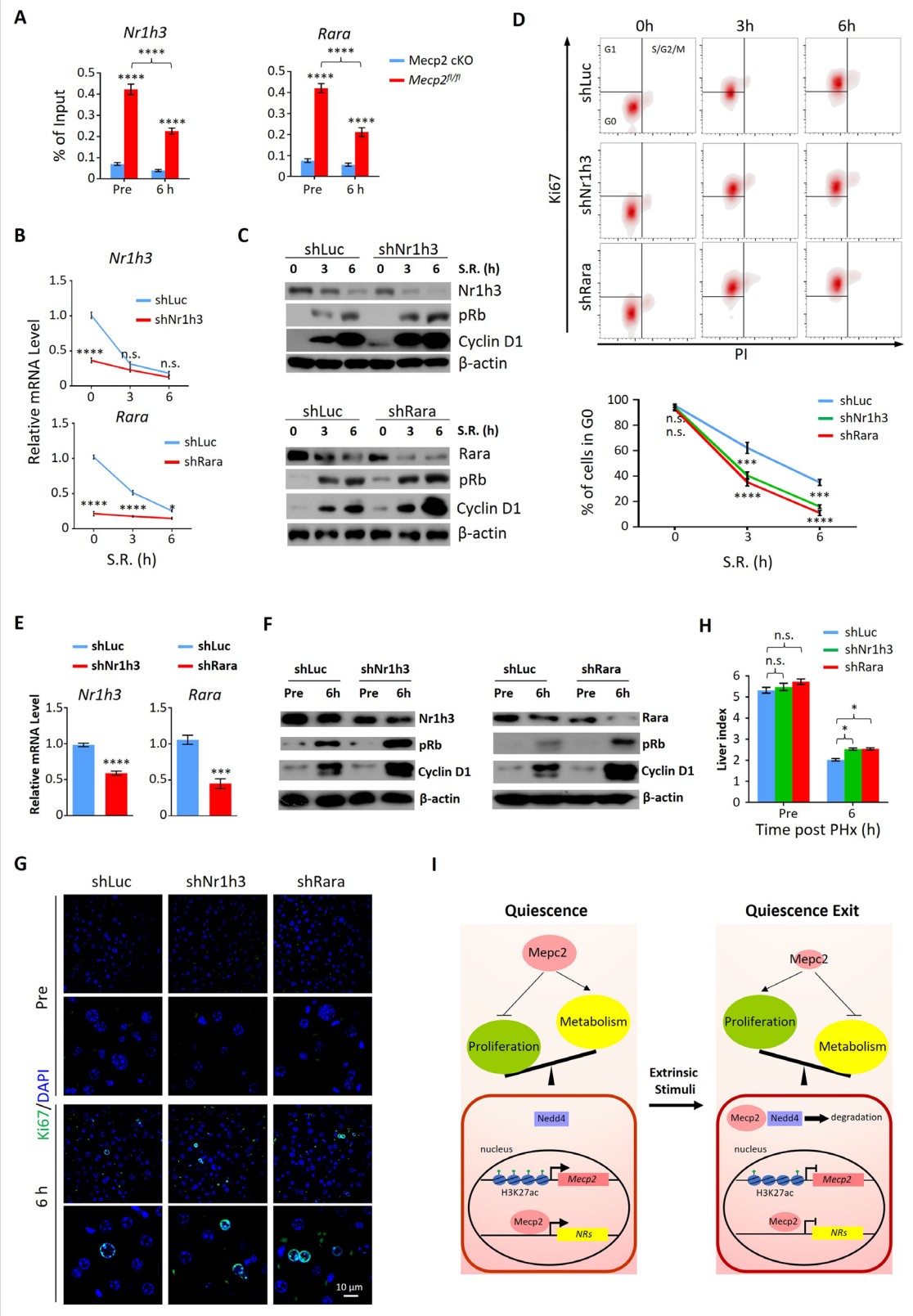

**Figure 7.** Depletion of either Nr1h3 or Rara mimics the Mecp2 knockdown (KD) phenotype during quiescence exit. (**A**) ChIP-qPCR analyses of Mecp2 at the promoter-proximal regions of *Rara* and *Nr1h3* in *Mecp2^fl/fl^* and Mecp2-cKO livers before and 6 hr post-partial hepatectomy (PHx). (**B–D**) Either Nr1h3 or Rara KD promotes serum restimulation (SR)-induced quiescence exit in 3T3 cells. (**B**) Real-time PCR showing lentivirus-mediated KD of either *Nr1h3* or *Rara*. shLuc served as a negative control. (**C**) Western blotting (WB) of pRb, Cyclin D1, Nr1h3, and Rara in control and Nr1h3 or Rara KD 3T3 cells at

*Figure 7 continued on next page*

*Figure 7 continued*

the indicated time points. (**D**) The effect of Nr1h3 or Rara KD on quiescent exit in 3T3 cells determined by Ki67/PI staining followed by flow cytometry. Data are presented as means ± SEM; In (**A, B**), n = 5; in (**D**), n = 3. *p<0.05; ***p<0.001; ****p<0.0001 by two-way ANOVA. (**E–H**) Either Nr1h3 or Rara KD further enhances quiescence exit in Mecp2 cKO livers. (**E**) Real-time PCR showing adeno-associated virus (AAV)-mediated KD of either Nr1h3 or Rara. shLuc served as a negative control. (**F**) WB of pRb, Cyclin D1, Nr1h3, and Rara in control and Nr1h3 or Rara KD 3T3 cells at the indicated time points. (**G**) The effect of Nr1h3 or Rara KD on quiescent exit in Mecp2 cKO livers determined by IF and liver index (**H**) before and 6 hr post-PHx. Data are presented as means ± SEM. In (**E, H**), n = 5 mice/group; n.s., not significant; *p<0.05, ****p<0.0001 by two-way ANOVA. (**I**) Model of the negative regulatory role for Mecp2 in fine-tuning quiescence exit.

The online version of this article includes the following source data for figure 7:

**Source data 1.** Depletion of either Nr1h3 or Rara mimics the Mecp2 KD phenotype during quiescence exit.

Despite 11 identified ubiquitination sites in Mecp2, our knowledge about the E3 ligases that catalyze the covalent attachment of ubiquitin to these sites remains scarce (*Lamonica et al., 2017*; *Bellini et al., 2014*). Recently, *Wang, 2014* uncovered the role of RING Finger Protein 4 (RNF4) in transcriptional activation by mediating the ubiquitination of Mecp2. Herein, we discovered Nedd4 as a novel regulator of Mecp2 protein stability. Nedd4 has recently been identified as an essential regulator of liver regeneration through post-translational modification of growth factor signaling (*Bachofner et al., 2017*). Consistent with our results, Bachofner et al. reported in vivo knockdown of Nedd4 in hepatocytes caused inhibition of cell proliferation after PHx. Although we identified Nedd4-mediated ubiquitination of Mecp2 in the cellular model of quiescence exit, we also measured the expression of Nedd4 at the early stages of liver regeneration (*Figure 5—figure supplement 1D*), which in turn may drive hepatocytes to enter the cell cycle by targeting Mecp2. Further studies are needed to explore what domain(s) is (are) responsible for the interaction between Mecp2 and Nedd4, which corresponding ubiquitylation sites within Mecp2 are targeted by Nedd4, and whether ubiquitylation site mutations of Mecp2 can completely abolish cell cycle reentry.

The switch from quiescence to active cycling requires coordinating all the necessary metabolic and cell cycle machinery. There is an urgent need to synthesize DNA, proteins, and lipids required for the generation of daughter cells in resting cells reentering the cell cycle (*Coller, 2019*). Regarding injury-induced hepatic regeneration, a global transcriptome shift from metabolism to proliferation is a reasonable strategy to satisfy the proliferating needs at the early stages of liver regeneration after PHx (*Liu and Chen, 2017*). Consistently, we observed an increase in gene expression involved in mRNA abundance, splicing, and translation, and a decrease in genes enriched in fatty acid, lipid, and amino acid metabolism at 6 hr post-PHx in a Mecp2-dependent manner (*Figure 6A and B*). This metabolic repression is transient and largely restored when the regenerating liver reaches its original size. Yet, how Mecp2 differentially regulates proliferation-associated and metabolic genes is a fascinating question that remains unanswered in our study and merits further investigation.

In this study, we discovered that Mecp2-mediated transcriptional activation of genes involved in metabolism is one of the mechanisms that prevents exit from quiescence. Specifically, we identified 10 NRs, which are ligand-dependent transcription factors that regulate cellular metabolism, proliferation, differentiation, and apoptosis. The NR superfamily can be divided into three classes based on their ligands and mechanisms of action, including the steroid receptor family, the thyroid/retinoid family, and the orphan receptor family (*Rudraiah et al., 2016*; *Pearen and Muscat, 2012*; *Zheng and Murphy, 2016*). To date, studies on NRs have elucidated the roles of several NRs in regulating hepatomegaly and liver regeneration, including peroxisome proliferator-activated receptors (PPARα or Nr1c1, and PPARγ or Nr1c3), pregnane X receptor (PXR, Nr1i2), constitutive androstane receptor (CAR, Nr1i3), liver X receptor (LXR, Nr1h3), and farnesoid X receptor (FXR, Nr1h4) (*Zhao et al., 2022*). Among these genes, *Nr1i3* and *Nr1h3* also emerged as PHx-repressed Mecp2-activated NRs in our study. Consistent with our observations, Sasso et al. demonstrated that Nr1h3, which is responsible for cholesterol catabolism and fatty acid synthesis, acts as an inhibitor of liver regeneration (*Lo Sasso et al., 2010*). However, they did not observe the decreased mRNA levels of both LXR isoforms (Nr1h3 and Nr1h2) mainly because they monitored the transcriptional levels at 1 d post-PHx, which was not early enough to capture the upstream changes in transcriptional regulation. In general, previous studies have barely focused on the connection between quiescence exit and NRs. Based on our observations, the inhibition of certain NRs by Mecp2 depletion during quiescence exit is probably general and not hepatocyte-specific because we also validated the repression of several NRs in cellular models

of quiescence exit. Additionally, the functional validation of Nr1h3 and Rara in 3T3 cells further supports the notion that Mecp2 may postpone cell cycle reentry through activating NRs. However, not all NRs exhibit the same function in quiescence exit. Huang and colleagues reported that the absence of the primary nuclear bile acid receptor FXR (also known as Nr1h4) strongly inhibited liver growth in the early stages of regeneration (*Huang et al., 2006*). Therefore, the cell type- and/or stimulus-specific function and detailed mechanisms for certain NRs in regulating quiescence exit await further investigation. Our study highlights the importance of NRs in mediating Mecp2-regulated quiescence exit, which may serve as attractive therapeutic targets after further investigation of the underlying mechanisms.

In summary, our study opens a brand-new perspective to understand the functional involvement of Mecp2 as a general regulator of quiescence exit and has provided insight into the mechanisms that may link metabolism to quiescence exit. Differential targeting of Mecp2 based on its different roles at different cell-cycle phases should be taken into consideration when examining potential clinical applications.

## Materials and methods

### Resources table

Detailed information on reagents, antibodies, primers, and shRNAs is listed in Appendix 1—key resources table.

### Experimental animals

All the animal studies were performed in accordance with the ethical guidelines of South Medical University ethics committee and were approved by the Ethics Committee on Use and Care of Animals of Southern Medical University (SMUL2017193). In this study, 10- to 12-week-old female C57BL/6 mice were purchased from the Laboratory Animal Centre of Southern Medical University (Guangzhou, China). The *Alb*-cre (JAX stock #025200) mouse strain was obtained from Jackson Laboratory (Bar Harbor, ME). *Mecp2^flox/flox^* (#NM-CKO-190001) mice were obtained from Shanghai Model Organisms Center (Shanghai, China). To generate hepatocyte-specific *Mecp2* knockout mice by the deletion of exons 2 and 3, we mated *Mecp2^flox/flox^* mice with *Alb*-Cre^+^ mice to obtain *Alb^+^Mecp2^flox/+^* female mice. The *Alb^+^Mecp2^flox/+^* female mice were then bred with *Mecp2^flox/Y^* male mice to obtain *Alb^+^Mecp2^flox/flox^* female mice (termed Mecp2 cKO) and littermate controls (*Alb^-^Mecp2^flox/flox^*). All mice were housed at 22°C under a 12 hr light/dark cycle. Food and water were provided ad libitum. Genotyping was carried out on tail DNA by polymerase chain reaction (PCR) using specific primers, and Mecp2 deficiency in hepatocytes was confirmed via WB and q-PCR. Littermate controls were used in experiments. The experiment was conducted in a random order using mice from different cages to minimize potential confounders such as the order of treatments and measurements and location of the animals and cages. As the experiment progressed, the experimenter became blind to the group allocation, and the principal investigator only knew the allocation at each stage.

### Partial hepatectomy surgery

For standard two-thirds PHx, 10- to 12-week-old mice were used in this study. Surgery was performed using the methodology described previously between 9:00 and 12:00 AM (*Mitchell and Willenbring, 2008*). The mice were then euthanized with pentobarbital at specified time points. To study the expression of Mecp2 in cell cycle progression, animals were killed with anesthetic overdose, and tissue was harvested at time points representing the G1 phase (6, 12, and 24 hr after PH), S/G2 phase (48 hr after PH), and the 'post-replicative' phase of liver regeneration (120 hr after PH). Livers from non-operated mice served as G0 phase (PHx 0 hr) controls. At the specified time points, livers were fixed in 10% buffered formalin for 24 hr or frozen in liquid nitrogen for later experiments. Liver and body weights were recorded at the time of death for calculating liver-to-body weight ratios.

### Cell culture and synchronization

NIH3T3 and HUVECs were obtained from American Type Culture Collection (Manassas, VA), and the HT22 cells were purchased from the Procell Life Science Technology (Wuhan, China). The cell lines were mycoplasma negative and authenticated by STR profiling. All cells were cultured in Dulbecco's

Modified Eagle Medium (DMEM) containing 4.5 g/l glucose and 10% fetal bovine serum (FBS). To synchronize cells in G0 by SS, cells were plated at a density of $1 \times 10^4$ cells per $cm^2$ overnight and allowed to attach to the tissue culture plate. Cells were washed three times with phosphate-buffered saline and starved in DMEM with 0.1% FBS for 30 hr (*Coller et al., 2006*). Then, the cells were induced into the cell cycle with 15% FBS and collected at the indicated times.

For cells arrested by CI, cells were plated at high density ($1 \times 10^5$ cells/$cm^2$), grown to confluence, and maintained at confluence for up to 3 d (*Wallbaum et al., 2009*). During this time, the cells undergo CI entering G0 arrest. The G0-phase cells were then plated at a density of $2 \times 10^4$ cells per $cm^2$ and cultured with 10% FBS. After attachment, cell samples were collected at 24, 48, and 72 hr. Subsequent analyses using qPCR, WB, IF, and cell cycle analysis were performed.

## FACS cell cycle profile analysis

Cell cycle phases were monitored by flow cytometry, as previously described (*Kim and Sederstrom, 2015*). The cells in different cell cycle phases were harvested and fixed overnight at 4°C with 70% ethanol. The following day, the fixed cells were centrifuged at $250 \times g$ for 5 min, and the pellet was resuspended in 1 ml PBS, centrifuged, and resuspended in 1 ml PI staining solution containing RNase A. After incubation at room temperature for 30 min in the dark, DNA content profiles were obtained via flow cytometry using a FACScan instrument. Gates were set over the G0/G1, S, and G2/M peaks, and then the percentages of cells in different cell cycle phases were calculated.

For Ki67 expression, PI-labeled cells were stained with Ki-67-APC antibody for 30 min at room temperature. Ki67 level and DNA content profiles were analyzed via flow cytometry. The percentage of cells in the G0 phase was defined as Ki-67⁻ and PI⁺ (2N) (*Kim and Sederstrom, 2015*). Flow cytometry data were analyzed with FlowJo V10 (FlowJo, USA).

## mRNA extraction and quantitative RT-PCR

Cells and liver samples were homogenized in 1.0 ml of TRIZOL. RNA was isolated using chloroform extraction and transcribed into cDNA using Prime Script Reverse Transcriptase at 1000 ng in a total volume of 20 µl following the manufacturer's protocol. A volume of 2 µl of cDNA was used as template for qPCR using SYBR Premix Ex Taq. qPCR reactions were performed using an ABI 7500 system. Samples were normalized for expression levels of human or mouse actin. The comparative ΔΔCt method was used to quantify the relative fold changes and β-actin mRNA served as an internal control. The specific primer sequences used are listed in Appendix 1—key resources table.

## RNA-seq data analysis

Total RNAs from control and Mecp2-KO mice at PHx 0 hr and PHx 6 hr were performed by Novogene (Beijing, China). The integrity of RNA was assessed using the RNA Nano 6000 Assay Kit of the Bioanalyzer 2100 system. After the fragmentation was carried out, first-strand cDNA was synthesized using random hexamer primer and M-MuLV Reverse Transcriptase, then used RNaseH to degrade the RNA. Second-strand cDNA synthesis was performed using DNA Polymerase I and dNTP. The library fragments were purified with AMPure XP system and then sequenced using the Illumina NovaSeq 6000. According to the Eoulsan pipeline (*Jourdren et al., 2012*), data were processed using read filters, mappings, alignment filters, read quantifications, normalizations, and differential analyses. Before mapping, polyN read tails were trimmed, reads ≤40 bases were removed, and reads with quality mean ≤ 30 were discarded. Alignments from reads matching more than once on the reference genome were removed using the Java version of Samtools (*Li et al., 2009*). All overlapping regions between alignments and referenced exons (or genes) were counted using HTSeq-count (*Pertea et al., 2016*). The normalization step and the differential analyses were carried out with DESeq2 (*Love et al., 2014*). RNA-seq reads were aligned to the mouse genome (mm10). p<0.05 and fold change ≥1.5 were set as the thresholds for significant differential expression.

## ChIP and ChIP-seq

ChIP assays were performed using the SimpleChIP Plus Enzymatic Chromatin IP Kit according to the manufacturer's procedures. Liver tissues were harvested at the indicated time and quickly cut into ~0.2 $cm^3$ pieces, crosslinked with 1% formaldehyde for 15 min, and quenched with 2.5 M glycine for 5 min at room temperature. After centrifugation, the pieces were homogenized in ice-cold PBS

first with a loose pestle and then with a tight pestle, permeabilized and resuspended in micrococcal nuclease (MNase) buffer, and incubated with MNase for 20 min at 37°C. The MNase-digested chromatin was centrifuged at 14,000 rpm for 1 min. Then, the cells were resuspended in ChIP buffer and the chromatin was sonicated on ice (30 s on/30 s off) to obtain soluble sheared chromatin (average DNA length of 150–450 bp). Approximately 10 μg of chromatin was diluted in ChIP buffer and pre-cleared with Dynabeads Protein G (Invitrogen, 10004D) for 2 hr at 4°C. Anti-H3K27ac antibodies and anti-Mecp2 antibodies were added to the pre-cleared chromatin, followed by rotation overnight at 4°C. The Dynabeads were washed with low/high salt solutions and eluted twice in TE buffer containing 1% SDS at 65°C for 15 min. The combined eluates were isolated by reversal of cross-linking, incubated with RNase A (10 μg/ml) followed by proteinase K (0.1 mg/ml), and the DNA was extracted using a phenol-chloroform extraction protocol. DNA quality was assessed using an Agilent bioanalyzer and quantified using a Qubit fluorometer.

Later, the immunoprecipitated chromatin DNA was sent to Novogene Company for ChIP-seq. Reads coming from Mecp2 were trimmed using Trimmomatic (*Bolger et al., 2014*) and aligned to the mouse genome (mm10) using Bowtie2 (*Langmead and Salzberg, 2012*), followed by processing using Samtools (*Li et al., 2009*). Peak calling was performed using MACS for narrow peaks (*Zhang et al., 2008*) and HOMER for broad peaks (*Heinz et al., 2010*), and peak annotation was performed using HOMER. Aligned reads were normalized using deepTools (*Ramírez et al., 2016*). By using the normalized BigWig files of ChIP and input samples, the averaged signal was quantified by BigWig Summary, and the ratio of ChIP/input was used as the ChIP signal intensity.

## ChIP-qPCR

H3K27ac and Mecp2 were assessed in promoters of genes of interest by ChIP-qPCR. ChIP-DNA from the liver was obtained as described above. Protein-bound DNA was amplified by qPCR. Data were then normalized to the input and expressed as fold changes (relative enrichment) compared with the control group. Normal rabbit IgG serve as the negative control. qPCR analyses of immunoprecipitated chromatin were performed for promoter sequences within +2.5 kb of the TSS of the analyzed genes. The percentage of input was then calculated for ChIP-qPCR signals. Promoter-specific primers used for these studies are listed in Appendix 1—key resources table.

## WB analysis

Mouse tissues or cell cultures were lysed in RIPA buffer (50 mM Tris–HCl pH 7.4, 1% NP-40, 0.25% Na-deoxycholate, 150 mM NaCl, 1 mM EDTA, pH 7.4) containing a cocktail of protease inhibitors and phosphatase inhibitors, followed by centrifugation at 10,000 × *g* for 10 min at 4°C. The supernatant protein quantity was determined using a BCA assay Equal amounts of protein (30 μg) were resolved by electrophoresis in a 10% or 12% gel and transferred to nitrocellulose membranes. The membranes were then incubated with specific antibodies. The membranes were then visualized by enhanced chemiluminescence using an ECL kit. Using ImageJ software, the values of the target protein/β-actin were calculated to evaluate the relative protein level.

## siRNA knockdown

We transiently transfected NIH3T3 cells with Mecp2 or Nedd4 siRNA (Genema, Shanghai, China) using Lipofectamine 3000 in Opti-MEM medium, according to the manufacturer's instructions. Transfected cells were arrested in G0 by 30 hr SS, then collected following 15% FBS stimulation for 3 or 6 hr. The effects of knockdown were evaluated by qPCR and WB. Subsequent analyses for IF and cell cycle analysis were also performed. The specific sequences used are listed in Appendix 1—key resources table.

## Gene overexpression experiments

NIH3T3 cells were transfected with Mecp2 or Nedd4 plasmid using Lipofectamine 3000 in serum-free medium for 12 hr, then the medium was removed and replaced with DMEM containing 10% FBS for 24 hr. Transfected cells were arrested in G0 for 30 hr by serum starvation, then collected followed by 15% FBS re-stimulation for 3 or 6 hr. Empty PCMV6-Entry vector was used as a control. The efficacy of overexpression was analyzed by RT-qPCR and WB analysis. Subsequent analyses for IF and cell cycle analysis were also performed.

## Histological staining and IHC

Murine liver biopsies were processed for histological analysis. The liver samples were fixed in 4% paraformaldehyde at room temperature for 24 hr, embedded in paraffin, and stained for histological analysis. After removal of paraffin, hematoxylin-eosin (H&E) staining was performed using a Hematoxylin and Eosin Staining Kit (Beyotime, Shanghai, China, #C0105S). For IHC staining, sections from liver biopsies were treated with citrate antigen retrieval solution for 3 min by high pressure. After blocking in 10% goat serum for 60 min at room temperature, the sections were processed for Mecp2 and Ki67 using diaminobenzidine according to the manufacturer's instructions.

## Immunofluorescence

To evaluate the expression of Mecp2 or Ki67 during the phase of liver generation, primary antibodies, including anti-Alb, anti-Mecp2, or anti-Ki67, were added to sections for 12 hr at 4°C. Sections were washed three times in PBS, followed by application of secondary antibody goat anti-mouse Alexa Fluor 488 or donkey anti-rabbit Alexa Fluor 594 at a 1:200 dilution for 1 hr at room temperature. Nuclei were counterstained using DAPI. Following a wash in PBS, tissues were mounted with 50% glycerol and viewed on a Nikon (Tokyo, JP) Eclipse epi-fluorescence microscope.

IF procedures were performed as follows: G0 and cell cycle re-entry NIH3T3 cells were fixed with 4% paraformaldehyde and then washed twice with PBS. Cells were then permeabilized with 0.2% Triton X-100 at 4°C for 15 min and subsequently blocked with 1% bovine serum albumin (BSA) for 60 min at room temperature. Primary antibodies included anti-Mecp2, anti-Nedd4, and anti-Ki67, which were incubated overnight at 4°C in the presence of 1% BSA. The cells were then visualized using secondary antibody conjugated to Alexa Fluor-488 or Alexa Fluor-594 as described above.

## Ubiquitination

To determine whether the Mecp2 protein was degraded by proteases during the phase of cell cycle reentry, NIH3T3 cells were synchronized by SS for 30 hr, collected following stimulation by 15% FBS with or without 10 μM MG132 for 3 or 6 hr, and subjected to WB. Endogenous protein in the Mecp2 ubiquitination assay was examined by IP. The cellular lysates of G0 and cell cycle reentry-cells were lysed in IP lysis buffer, and the supernatant was obtained by centrifugation at 10,000 × g for 10 min at 4°C. Then, 1 mg of total protein was incubated with anti-Mecp2 (5 μg) antibody overnight at 4°C with constant mixing. Antigen-antibody complexes were incubated with magnetic beads for 2 hr with shaking. After three washings, retained proteins were eluted using 30 μl of SDS lysis buffer. Eluted Mecp2-associated cellular proteins were separated by SDS-PAGE. Ubiquitination was analyzed using anti-ubiquitin antibody.

## Co-IP assay

NIH3T3 cells were synchronized in G0 phase by SS as previously described, followed by 15% FBS stimulation for 3 or 6 hr. Cells were washed with cold PBS and lysed in IP lysis buffer supplemented with protease inhibitors. The supernatant was obtained by centrifugation at 10,000 × g for 10 min at 4 °C. Protein concentrations were quantified using a BCA Protein Assay Kit. Then, 1 mg of total protein was incubated with anti-Mecp2 (5 μg) or anti-Nedd4 (5 μg) antibody overnight at 4 °C with constant mixing. Also, 30 μl of Dynabeads were added and incubation was continued for an additional 2 hr. After three washings with PBS, retained proteins were eluted using 30 μl of SDS lysis buffer. Protein complexes were then detected by WB and immunoblotted with anti-Mecp2 or Nedd4 antibodies.

## Mass spectrometry assay

To reveal the proteins of the ubiquitination system possibly interacting with Mecp2, NIH3T3 cells were synchronized by SS for 36 hr and collected following 15% FBS stimulation for 3 or 9 hr. Total protein (1 mg) was incubated with anti-Mecp2 (5 μg) overnight at 4°C with constant mixing. IgG was used as the negative control. Then, antibody was incubated with Dynabeads for 2 hr with shaking. After three washings, retained proteins were eluted using 30 μl of SDS lysis buffer. Eluted Mecp2-associated cellular proteins were separated by SDS-PAGE and stained with Coomassie blue. Trypsin was used to digest stained protein bands. An Orbitrap Elite mass spectrometer was used to analyze the digested samples by Applied Protein Technology Co., Ltd (Shanghai, China). Using Mascot as

a search engine, fragment spectra were scanned against the UniProt database to identify proteins. In this assay, Nedd4 was identified as the most abundant peptide of E3 ligase during the G0-G1 transition.

## Lentiviral vector constructs and transduction

For stable knockdown of Nr1h3 and Rara, lentiviruses were generated according to our previous protocol (*Cao et al., 2022*). Briefly, shRNAs targeting luciferase or mouse Nr1h3 and Rara were cloned into the pLKO.1 vector. The lentiviral vectors were co-transfected with the packaging vectors pCMV-deltaR8 and pCMV-VsVg into LentiX-293T cells to generate virus. After 48 hr, virus was collected and used to infect NIH3T3 cells with 6 µg/ml polybrene for another 12 hr. Infected cells were selected in puromycin for 3 d, and the expression of Nr1h3 and Rara in infected cells was verified by qRT-PCR. We used pLKO.1-luciferase-Puro empty vector as a negative control. The sequences of the shRNAs are listed in Appendix 1—key resources table.

## AAV production and tail vein injection

In vivo Mecp2 overexpression was achieved by recombinant adeno-associated virus serotype 8 (AAV8) vectors. AAV8 vectors carrying Mecp2 or GFP sequences with a thyroxin-binding globulin (TBG) promoter (AAV8-TBG-GFP, AAV8-TBG-Mecp2) were manufactured by Genechem Co., Ltd (Shanghai, China). AAV8-TBG-GFP served as negative control. AAV8-TBG-GFP/Mecp2 vectors ($2 \times 10^{11}$ vector genomes per mouse) were injected intravenously into C57 mice (termed AAV-Mecp2 mice) or CKO$^{Alb-Mecp2}$ mice (termed Mecp2 cKO/AAV-Mecp2 mice), respectively. After 4 wk, 70% PHx was performed as described above. Mecp2 overexpression were verified by IF and WB. The remaining livers were collected at 6 and 48 hr after surgery. Liver-to-body weight ratios were calculated as described above.

To knock down the expression of Nr1h3 or Rara in Mecp2-KO mice, AAV8-mediated delivery of shRNAs was used in this study. The vehicle vector ssAAV-TBG-mNeonGreen-WPRE-SV40pA was used as a negative control (termed shLuc). AAV production was performed according to our previous method (*Cao et al., 2022*). Briefly, 293T cells were co-transfected with various plasmids. We transfected 40 µg of total DNA (5.7 µg of pAAV8, 11.4 µg of pHelper, and 22.8 µg of TBG-NeonGreen-mir30-shNr1h3 [termed shNr1h3] or 22.8 µg of TBG-NeonGreen-mir30-shRara [termed shRara]) into 293T cells in a 15 cm dish. After 12 hr, the transfection medium was changed to normal medium. The medium containing AAV was collected at 72 and 120 hr post-transfection, and the cells were collected at 120 hr post-transfection. AAV particles were digested from cells by salt-active nuclease. Subsequently, AAV vector particles were purified by ultracentrifugation in an iodixanol density gradient at $350,000 \times g$ for 2 hr at 18°C. The virus titer was determined by real-time PCR. AAV vector ($2 \times 10^{11}$ vector genomes per mouse) were injected intravenously via the tail vein to Mecp2 cKO mice. After 4 wk, 70% PHx was performed as described above. The knockdown efficacy of Nr1h3 and Rara was verified by qPCR. The remaining livers were collected 6 hr after surgery. Liver-to-body weight ratios were calculated as described above.

## Accession numbers

ChIP-seq and mRNA-seq data were submitted to the GEO repository under accession numbers GSE227727 and GSE227723. The mass spectrometry proteomics data were deposited to the *ProteomeXchange* Consortium and are available via *ProteomeXchange* (PXD042085).

## Statistics

We did not use any particular methods to determine whether the data met the statistical assumptions. Statistical significance in each group was analyzed by Student's *t*-test, one-way ANOVA, or two-way ANOVA.

## Acknowledgements

This work was supported by grants from the National Natural Science Foundation of China (31900507, 92268204), the Guangdong Basic and Applied Basic Research Foundation (2019A1515011511), and the China Postdoctoral Science Foundation (2019M652955, 2023M731541).

## Additional information

### Funding

| Funder | Grant reference number | Author |
| --- | --- | --- |
| National Natural Science Foundation of China | 31900507 | Jun Yang |
| National Natural Science Foundation of China | 92268204 | Xiaochun Bai |
| Guangdong Provincial Applied Science and Technology Research and Development Program | 2019A1515011511 | Jun Yang |
| China Postdoctoral Science Foundation | 2019M652955 | Jun Yang |
| China Postdoctoral Science Foundation | 2023M731541 | Jun Yang |

The funders had no role in study design, data collection and interpretation, or the decision to submit the work for publication.

### Author contributions

Jun Yang, Conceptualization, Data curation, Software, Formal analysis, Funding acquisition, Validation, Investigation, Methodology, Writing – original draft, Project administration, Writing - review and editing; Shitian Zou, Conceptualization, Data curation, Project administration; Zeyou Qiu, Conceptualization, Data curation, Software, Formal analysis, Visualization; Mingqiang Lai, Qing Long, Ping lin Lai, Data curation, Formal analysis; Huan Chen, Data curation, Methodology; Sheng Zhang, Zhi Rao, Xiaoling Xie, Yan Gong, Resources; Anling Liu, Resources, Writing – original draft; Mangmang Li, Resources, Visualization, Methodology, Writing – original draft, Project administration; Xiaochun Bai, Supervision, Funding acquisition, Investigation, Writing – original draft

### Author ORCIDs

Jun Yang http://orcid.org/0000-0002-7223-6587
Xiaochun Bai http://orcid.org/0000-0001-9631-4781

### Ethics

All the animal studies were performed in accordance with the ethical guidelines of South Medical University ethics committee and were approved by the Ethics Committee on Use and Care of Animals of Southern Medical University (SMUL2017193).

Reviewer #1 (Public Review): https://doi.org/10.7554/eLife.89912.3.sa1
Author response https://doi.org/10.7554/eLife.89912.3.sa2

## Additional files

### Supplementary files

• Supplementary file 1. For the GO enrichment analysis, the binding proteins of Mecp2 during the cell cycle reentry.
• MDAR checklist

### Data availability

ChIP-seq and mRNA-seq data have been submitted to the GEO repository under accession number GSE227727 and GSE227723. The mass spectrometry proteomics data have been deposited to the ProteomeXchange Consortium and are available via ProteomeXchange (PXD042085). All data generated or analysed during this study are included in the manuscript and supporting files; source data files have been provided.

The following datasets were generated:

| Author(s) | Year | Dataset title | Dataset URL | Database and Identifier |
|-----------|------|---------------|-------------|-------------------------|
| Yang J, Qiu Z | 2024 | Mecp2 fine-tunes quiescence exit by targeting nuclear receptors [RNA-Seq] | https://www.ncbi.nlm.nih.gov/geo/query/acc.cgi?acc=GSE227723 | NCBI Gene Expression Omnibus, GSE227723 |
| Yang J, Qiu Z | 2024 | Mecp2 fine-tunes quiescence exit by targeting nuclear receptors [ChIP-Seq] | https://www.ncbi.nlm.nih.gov/geo/query/acc.cgi?acc=GSE227727 | NCBI Gene Expression Omnibus, GSE227727 |
| Yang J, Qiu Z | 2024 | IP-mass spectrometry of MeCP2 binding proteins during cell cycle re-entry | https://proteomecentral.proteomexchange.org/cgi/GetDataset?ID=PXD042085 | ProteomeXchange, PXD042085 |

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

# Appendix 1

## Appendix 1—key resources table

| Reagent type (species) or resource | Designation | Source or reference | Identifiers | Additional information |
| --- | --- | --- | --- | --- |
| Cell line (*Homo sapiens*) | HUVEC | ATCC | Cat# CRL-1730 | |
| Cell line (*H. sapiens*) | LentiX-293T | ATCC | Cat# ACS-4500 | |
| Cell line (*Mus musculus*) | NIH3T3 | ATCC | Cat# SCSP-515 | |
| Cell line (*M. musculus*) | Ht22 | Procell Life Science&Technology | Cat# CL-0697 | |
| Antibody | Albumin (mouse monoclonal) | Proteintech | Cat# 66051-1-Ig; RRID:AB_11042320 | 1:100 |
| Antibody | β-Actin (mouse monoclonal) | Proteintech | Cat# 66009-1-Ig; RRID:AB_2687938 | 1:4000 |
| Antibody | Cyclin A2 (mouse monoclonal) | abcam | Cat# ab38; RRID:AB_304084 | 1:1000 |
| Antibody | Cyclin B1 (mouse monoclonal) | abcam | Cat# ab72; RRID:AB_305751 | 1:1000 |
| Antibody | Cyclin D1 (rabbit monoclonal) | abcam | Cat# ab16663; RRID:AB_443423 | 1:1000 |
| Antibody | Cyclin E1 (rabbit monoclonal) | Cell Signaling Technology | Cat# 20808; RRID:AB_2783554 | 1:1000 |
| Antibody | H3K27ac (rabbit polyclonal) | abcam | Cat# ab4729; RRID:AB_2118291 | 1:50 |
| Antibody | IgG (rabbit, IgG) | Cell Signaling Technology | Cat# 2729; RRID:AB_1031062 | 1:50 |
| Antibody | K48-ubiquitin (rabbit monoclonal) | abcam | Cat# ab140601 | 1:1000 |
| Antibody | Ki67-Immunofluorescence (mouse monoclonal) | abcam | Cat# ab279653 | 1:100 |
| Antibody | Ki67-FACS (rat monoclonal) | BioLegend | Cat# 652406; RRID:AB_2561930 | 1:100 |
| Antibody | MeCP2 (rabbit monoclonal) | Cell Signaling Technology | Cat# 3456; RRID:AB_2143849 | 1:1000 |
| Antibody | MeCP2-ChIP (rabbit polyclonal) | abcam | Cat# ab2828; RRID:AB_2143853 | 1:50 |
| Antibody | Nedd4 (rabbit polyclonal) | Proteintech | Cat# 21698-1-AP; RRID:AB_10858626 | 1:1000 |
| Antibody | Nr1h3 (rabbit polyclonal) | Proteintech | Cat# 14351-1-AP; RRID:AB_10640525 | 1:1000 |
| Antibody | Rara (rabbit polyclonal) | Proteintech | Cat# 10331-1-AP; RRID:AB_2177742 | 1:1000 |
| Antibody | p-Rb S807/811 (rabbit monoclonal) | Cell Signaling Technology | Cat# 8516; RRID:AB_11178658 | 1:1000 |
| Antibody | Ubiquitin (mouse monoclonal) | Cell Signaling Technology | Cat# 3936; RRID:AB_331292 | 1:1000 |
| Antibody | Donkey anti-Mouse IgG (H+L) Highly Cross-Adsorbed Secondary Antibody, Alexa Fluor 488 (mouse polyclonal) | Thermo Fisher Scientific | Cat# A21202; RRID:AB_141607 | 1:100 |
| Antibody | Donkey anti-Rabbit IgG (H+L) Highly Cross-Adsorbed Secondary Antibody, Alexa Fluor 594 (rabbit polyclonal) | Thermo Fisher Scientific | Cat# A21207; RRID:AB_141637 | 1:100 |
| Recombinant DNA reagent | pLKO.1 vector | Addgene | Cat #8453; RRID:Addgene_8453 | |
| Recombinant DNA reagent | pCMV-deltaR8 | Addgene | Cat #12263; RRID:Addgene_12263 | |
| Recombinant DNA reagent | pCMV-VsVg | Addgene | Cat #8454; RRID:Addgene_8454 | |
| Recombinant DNA reagent | AAV8-TBG-MeCP2 | Genechem Co., Ltd | GOSV0233517 | |
| Recombinant DNA reagent | Mecp2 Mouse Tagged ORF Clone, transcript variant 1 | OriGene | Cat# MR226839 | |
| Recombinant DNA reagent | Mecp2 Mouse Tagged ORF Clone, transcript variant 2 | OriGene | Cat# MR207745 | |
| Recombinant DNA reagent | Nedd4 Mouse Tagged ORF Clone | OriGene | Cat# MR222243 | |

*Appendix 1 Continued on next page*

*Appendix 1 Continued*

| Reagent type (species) or resource | Designation | Source or reference | Identifiers | Additional information |
|---|---|---|---|---|
| Recombinant DNA reagent | pCMV6-Entry Mammalian Expression Vector | OriGene | Cat# PS100001 | |
| Chemical compound, drug | MG132 | selleck | Cat# S2619 | 10 µM |
| Chemical compound, drug | Puromycin | Sigma-Aldrich | Cat# P8833 | 2 µg/ml |
| Chemical compound, drug | Polybrene | Sigma-Aldrich | Cat# TR-1003 | 6 µg/ml |
| Chemical compound, drug | Pen Strep | Gibco | Cat# 15140-122 | 1% |
| Commercial assay or kit | Complete Protease Inhibitor mini EASY packs EDTA-Free | Roche | Cat# 05892791001 | |
| Commercial assay or kit | Lipofectamine 3000 Transfection Reagent | Thermo Fisher Scientific | Cat# L3000-015 | |
| Commercial assay or kit | Propidium Iodide (PI)/RNase Staining Solution | Cell Signaling Technology | Cat# 4087 | |
| Commercial assay or kit | DAPI | Thermo Fisher Scientific | Cat# D-1306 | |
| Commercial assay or kit | Pierce IP Lysis Buffer | Thermo Fisher Scientific | Cat# 87787 | |
| Commercial assay or kit | DAB Kit | ZSGB-BIO | Cat# ZLI-9018 | |
| Commercial assay or kit | Western Lightning Plus ECL | PerkinElmer | Cat# 0RT2655 | |
| Commercial assay or kit | SimpleChIP Plus Enzymatic Chromatin IP Kit | Cell Signaling Technology | Cat# 9005 | |
| Commercial assay or kit | Dynabeads Protein G | Thermo Fisher Scientific | Cat# 10004D | |
| Commercial assay or kit | Coomassie Blue Super Fast Staining Solution | Beyotime | Cat# P0017F | |
| Sequence-based reagent | siRNA to MeCP2 #1 | This paper | Suzhou GenePharma Co., Ltd | CCUGAAGGUUGGACACGAA |
| Sequence-based reagent | siRNA to MeCP2 #2 | This paper | Suzhou GenePharma Co., Ltd | UGACAAAGCUUCCCGAUUA |
| Sequence-based reagent | siRNA to MeCP2 #3 | This paper | Suzhou GenePharma Co., Ltd | CCGAAUUGCUGCUGCUUUA |
| Sequence-based reagent | siRNA to MeCP2 #4 | This paper | Suzhou GenePharma Co., Ltd | CGAAAUGGCUGUGUAGCAA |
| Sequence-based reagent | siRNA to Nedd4: #1 | This paper | Suzhou GenePharma Co., Ltd | CAGUGAUCCUUACGUAAGATT |
| Sequence-based reagent | siRNA to Nedd4 #2 | This paper | Suzhou GenePharma Co., Ltd | GGGAAAUCGUACGAGAAGATT |
| Sequence-based reagent | siRNA to Nedd4 #3 | This paper | Suzhou GenePharma Co., Ltd | GGAGGAUUAUGGGUGUGAATT |
| Sequence-based reagent | Alb, F | This paper | PCR primer | ACCTGAAGATGTTCGCGATTATCT |
| Sequence-based reagent | Alb, R | This paper | PCR primer | ACCGTCAGTACGTGAGATATCTT |
| Sequence-based reagent | MeCP2, F | This paper | PCR primer | GCTGGGGCCCTTGTTTTGAAT |
| Sequence-based reagent | MeCP2, R | This paper | PCR primer | GCTTTAGGTTGCTGGTGATA |
| Sequence-based reagent | Ar, F | This paper | PCR primer | CTGGGAAGGGTCTACCCAC |
| Sequence-based reagent | Ar, R | This paper | PCR primer | GGTGCTATGTTAGCGGCCTC |
| Sequence-based reagent | α-actin mouse, F | This paper | PCR primer | GGCTGTATTCCCCTCCATCG |
| Sequence-based reagent | α-actin mouse, R | This paper | PCR primer | CCAGTTGGTAACAATGCCATGT |
| Sequence-based reagent | α-actin human, F | This paper | PCR primer | CACCATTGGCAATGAGCGGTTC |
| Sequence-based reagent | α-actin human, R | This paper | PCR primer | AGGTCTTTGCGGATGTCCACGT |
| Sequence-based reagent | MeCP2 common, F | This paper | PCR primer | TATTTGATCAATCCCCAGGG |
| Sequence-based reagent | MeCP2 common, R | This paper | PCR primer | CTCCCTCTCCCAGTTACCGT |
| Sequence-based reagent | MeCP2 mouse, F | This paper | PCR primer | GAGCGGCACTGGGAGACC |
| Sequence-based reagent | MeCP2 mouse, R | This paper | PCR primer | CTGGATGGTGGTGATGAT |
| Sequence-based reagent | MeCP2 human, F | This paper | PCR primer | GATGTGTATTTGATCAATCCC |
| Sequence-based reagent | MeCP2 human, R | This paper | PCR primer | TTAGGGTCCAGGGATGTGTC |
| Sequence-based reagent | Nedd4, F | This paper | PCR primer | TCGGAGGACGAGGTATGGG |

*Appendix 1 Continued on next page*

*Appendix 1 Continued*

| Reagent type (species) or resource | Designation | Source or reference | Identifiers | Additional information |
|---|---|---|---|---|
| Sequence-based reagent | Nedd4, R | This paper | PCR primer | GGTACGGATCAGCAGTGAACA |
| Sequence-based reagent | Nr1h3, F | This paper | PCR primer | CTCAATGCCTGATGTTTCTCCT |
| Sequence-based reagent | Nr1h3, R | This paper | PCR primer | TCCAACCCTATCCCTAAAGCAA |
| Sequence-based reagent | Nr1i3, F | This paper | PCR primer | ATATGGGCCGAGGAACTGTGT |
| Sequence-based reagent | Nr1i3, R | This paper | PCR primer | GGCGTGGAAATGATAGCCTGT |
| Sequence-based reagent | Nr2f6, F | This paper | PCR primer | GAGGACGATTCGGCGTCAC |
| Sequence-based reagent | Nr2f6, R | This paper | PCR primer | GTAATGCTTTCCACTGGACTTGT |
| Sequence-based reagent | Nr3c1, F | This paper | PCR primer | AGCTCCCCCTGGTAGAGAC |
| Sequence-based reagent | Nr3c1, R | This paper | PCR primer | GGTGAAGACGCAGAAACCTTG |
| Sequence-based reagent | Nr4a1, F | This paper | PCR primer | TTGAGTTCGGCAAGCCTACC |
| Sequence-based reagent | Nr4a1, R | This paper | PCR primer | GTGTACCCGTCCATGAAGGTG |
| Sequence-based reagent | Nr5a2, F | This paper | PCR primer | TGAGGAACAACTCCGGGAAAA |
| Sequence-based reagent | Nr5a2, R | This paper | PCR primer | CAGACACTTTATCGCCACACA |
| Sequence-based reagent | Nr6a1, F | This paper | PCR primer | CGCAACGGTTTCTGTCAGGAT |
| Sequence-based reagent | Nr6a1, R | This paper | PCR primer | GTTCAGCTCGATCATCTGGGA |
| Sequence-based reagent | Ppard, F | This paper | PCR primer | CTCATGAATGTGCCCCAGGT |
| Sequence-based reagent | Ppard, R | This paper | PCR primer | GTGCAGCAAGGTCTCACTCT |
| Sequence-based reagent | Rxrg, F | This paper | PCR primer | CATGAGCCCTTCAGTAGCCTT |
| Sequence-based reagent | Rxrg, R | This paper | PCR primer | CGGAGAGCCAAGAGCATTGAG |
| Sequence-based reagent | Rara, F | This paper | PCR primer | ATGTACGAGAGTGTGGAAGTCG |
| Sequence-based reagent | Rara, Reserve | This paper | PCR primer | ACAGGCCCGGTTCTGGTTA |
| Sequence-based reagent | Srebf1, F | This paper | PCR primer | GCAGCCACCATCTAGCCTG |
| Sequence-based reagent | Srebf1, R | This paper | PCR primer | CAGCAGTGAGTCTGCCTTGAT |
| Sequence-based reagent | MeCP2, F | This paper | ChIP-qPCR primer | TAAGTGACAGGAGTCACAGCG |
| Sequence-based reagent | MeCP2, R | This paper | ChIP-qPCR primer | TGGGACGTTGTATGTAACGGG |
| Sequence-based reagent | Nr1h3, F | This paper | ChIP-qPCR primer | CAGCACGTTGTAATGGAAGCC |
| Sequence-based reagent | Nr1h3, R | This paper | ChIP-qPCR primer | TAGCATTCAGTGGAGGGAAGG |
| Sequence-based reagent | Rara, F | This paper | ChIP-qPCR primer | CGATGAGTGGCAAGGTCTTT |
| Sequence-based reagent | Rara, R | This paper | ChIP-qPCR primer | ATAGCATAGCACCAGGGACAC |
| Sequence-based reagent | shRNA for Nr1h3 | This paper | shRNA sequence | CCTCAAGGACTTCAGTTACAA |
| Sequence-based reagent | shRNA for Rara | This paper | shRNA sequence | GAGCAGCAGTTCCGAAGAGAT |
| Other | B6. Cg-Speer6-ps1Tg (Alb-cre)21Mgn/J mice | The Jackson Laboratory | Cat# 003574 | Hepatocyte-specific Cre-transgenic mouse |
| Other | *MeCP2flox/flox* | Shanghai Model Organisms Center | Cat #NM-CKO-190001 | Mice carrying the targeted *MeCP2* allele |
| Other | C57/BL6 | Guangdong Medical Laboratory Animal Center | Cat #17 | Wild-type mouse |
| Software, algorithm | GraphPad Prism | GraphPad Software | Version 8.0 | |
| Software, algorithm | ModFit | ModFit LT Software | Version 4.1 | https://www.vsh.com/products/mflt/index.asp |
| Software, algorithm | ImageJ | ImageJ software | | https://imagej.nih.gov/ij/ |

