## [Editor Report · eLife assessment]

This **fundamental** study provides insights into the mechanism controlling cell cycle reentry, establishing a regulatory role for Mecp2 degradation in shifting transcription from metabolic to proliferation genes during quiescence exit. The evidence, which includes experimental data from in vitro cell culture and an in vivo injury-induced liver regeneration model, is **convincing** but the trigger for MeCP2 degradation and how MeCP2 differentially regulates proliferation and metabolic genes remains unclear.

---

## [Referee Report · Reviewer #1 (Public Review)]

In the study described in the manuscript, the authors identified Mecp2, a methyl-CpG binding protein, as a key regulator involved in the transcriptional shift during the exit of quiescent cells into the cell cycle. Their data show that Mecp2 levels were remarkably reduced during the priming/initiation stage of partial hepatectomy-induced liver regeneration and that altered Mecp2 expression affected the quiescence exit. Additionally, the authors identified Nedd4 E3 ligase that is required for downregulation of Mecp2 during quiescence exit. This is an interesting study with well-presented data that supports the authors' conclusions regarding the role of Mecp2 in transcription regulation during the G0/G1 transition. However, the significance of the study is limited by a lack of mechanistic insights into the function of Mecp2 in the process. This weakness can be addressed by identifying the signaling pathway(s) that trigger Mecp2 degradation during the quiescence exit.

---

## [Author Response]

The following is the authors’ response to the original reviews.

**Reviewer #1 (Recommendations for The Authors):**
(1) Since the data suggests that the degradation of Mecp2 is a crucial event in the exit from quiescence, gaining a better understanding of the underlying mechanism would improve the significance of the study. In this regard, the authors should take advantage of the serum stimulated degradation of Mecp2 (Fig. 3D) to identify the signaling pathway(s) required for the degradation.

Thank you for this suggestion. To decipher the molecular mechanisms underlying Mecp2-regulated quiescence exit, we performed RNA-seq combined with ChIP-seq to identify the Mecp2-dependent transcriptome genome-wide during the early stage of liver regeneration (Figure S6C). There were 2658 Mecp2 direct target genes, in which 537 were PHx-activated and 2121 were PHx-repressed genes (Figure 6A). GO analysis showed that PHx-activated Mecp2 targets were highly enriched in proliferation-associated biological processes such as ribosome biogenesis, rRNA metabolic process, ncRNA metabolic process, and regulation of transcription by RNA polymerase I, whereas PHx-repressed Mecp2 targets were associated with several metabolic processes including carboxylic acid catabolic process, cellular amino acid metabolic process, fatty acid metabolic process and steroid metabolic process (Figure 6B). These results suggest that Mecp2 plays a negative regulatory role during quiescence exit by activating metabolism-associated genes while repressing proliferation-associated genes in quiescent cells.

Given the more rapid decay of Mecp2 at the protein compared to the mRNA level during the quiescence-proliferation transition, we speculated that Mecp2 is targeted by posttranslational regulation. This hypothesis was supported by proteasome inhibition with the proteasome inhibitor MG132, which attenuated the reduction of Mecp2 in quiescent cells after S.R. (Figure S5A). To identify the signaling pathway that regulate Mecp2 degradation during the G0/G1 transition, we performed immunoprecipitation followed by mass spectrometry (IP-MS) using Mecp2 antibody in quiescent 3T3 cells treated with or without S.R. (Figure S5B). A total of 647 proteins were identified as putative Mecp2 interactors. We were particularly interested in the proteins involved in proteasome-mediated ubiquitin-dependent protein catabolic process which was one of the enriched Gene Ontology (GO) items in the Mecp2 interactome (Table S1).

(2) The authors suggest that Mecp2 downregulation accelerates the induction of pRb, which serves as a key marker for G0/G1 transition. However, their data only show increased magnitudes of the expression in Mecp2 downregulated cells at the timepoints when samples were collected (Figs. 2B and 4B). In the in vitro experiments, the authors should investigate earlier timepoints to demonstrate that induction of pRB during the quiescence exit occurs earlier in Mecp2 deficient cells compared to control cells. Likewise, a later induction of pRB in Mecp2 overexpression cells, in comparison to normal cells, should be demonstrated.

Thank you for these valuable suggestions. We have, accordingly, collected cell samples re-entered the cell cycle at 30-, 60-, 90- and 120-minutes post-S.R. We examined the pRb expression and found that phosphorylation of retinoblastoma protein (pRb) at Ser807/811 occurs earlier (about 90 minutes) in Mecp2 deficient cells compared to control cells (Figure S4C). Compared to the EV, Mecp2 OE resulted in the delayed induction of pRB (about 60 minutes) upon S.R. (Figure S4D). These data indicate that enhanced reduction of Mecp2 stimulates exit from quiescence.

(3) There are three well-known phosphorylation sites in Mecp2, including S80, S229, and S423. As protein ubiquitination and degradation are often triggered by phosphorylation, it would be interesting to examine whether phosphorylation at these sites of Mecp2 is required for its downregulation during quiescence exit. This can be achieved using non-phosphorylate mutants of Mecp2.

This is a very good question. Indeed, the 26S ubiquitin-proteasome system (26S UPS) is responsible for the breakdown of MeCP2 (PMID: 28394263, 28973632). In 2009, the bona fide PEST (enriched in proline, glutamic acid, serine, and threonine) domains have been identified, which are highly conserved across vertebrate evolution (PMID: 19319913). Consensus sequences enriched in PEST residues have been found to predispose proteins containing them for rapid proteolytic degradation (PMID: 8755249, 2876518). In addition, phosphorylation within PEST motifs precedes ubiquitination of proteins (PMID: 15229225). One of the best characterized sites of MeCP2 phosphorylation (S80) (PMID: 19225110), as well as one of the identified ubiquitination sites (K82/K99) (PMID: 22615490), both fall within one of these regions. It is still noteworthy that most of the MeCP2 phosphorylation sites were found in close proximity to potential ubiquitylation sites. For example, Rett syndrome missense mutations in Rett syndrome affecting three (K82R, K135A, K256S) of the ubiquitination sites (PMID: 25165434) and S80 (within one of the PEST sequences) and K82 have been shown to be phosphorylated and ubiquitinated.

Based on the above discussion, we providing a potential hypothesis that the MeCP2 turnover during cell cycle re-entry is achieved by an initial phosphorylation signal (phosphorylated at S80, S229, or S421) that triggers the ubiquitination of a close lysine residue. We hope to solve these issues and be able to present the findings in future work. Thanks again for your professional suggestions.

(4) It would be interesting if the authors could also examine the effect of altered expression of Mecp2 on the maintenance of quiescence. For example, whether the downregulation of Mecp2 sensitizes quiescent cells for entry of the cell cycle in response to serum stimulation or delays withdrawal from the cell cycle upon serum starvation or contact inhibition.

Thank you for your suggestions. Cell cycle synchronization was induced with serum deprivation. When nutrients are exhausted, altered expression of Mecp2 have no statistical influence on the maintenance of quiescence as analyzed by Flow cytometric (Figure 4D and H). This suggests that the altered expression of Mecp2 alone may not be sufficient for cell cycle exit. In the presence of growth factors or nutrients, loss of MeCP2 only accelerates the rate of cell cycle re-entry.

Minor points:For Figs. 2D, 2H, and 2L, it would be more intuitive if the percentage of changes in liver index rather than the relative index values were used. Also, the values listed in the figures should start from time zero after partial hepatectomy rather than pre-surgery.

Liver weight have the corresponding change with body weight. The liver index (ratio of regenerate liver weight/body weight) is tightly regulated and depends on metabolic demands of the organism. During the course of liver regeneration, reestablishment of liver volume after resection is regulated by the functional needs of the organism. Using the percentage of regenerate liver weight/body weight as a liver growth index could reflect the regenerative function.Next, we agree with the data presentation form and the values listed in the figures have been modified in the revised version.

**Reviewer #2 (Recommendations for The Authors):**
My concerns are as follows:(1) The authors note that the decrease in Mecp2 protein levels was more pronounced than the decrease in mRNA levels, suggesting the presence of post-translational regulation of Mecp2 during the early stages of G0 exit. Could the decrease in MeCP2 levels be related to autophagy flux?

Thank you for your valuable comments. Also, we have compared the cells extracts from untreated and chloroquine-treated cells (to block lysosomal degradation). Chloroquine did not cause any accumulation of MeCP2 (Figure S5B). The results suggest that autophagy activity do not involve in the decrease the MeCP2 protein.

(2) In addition to Cyclin D1, how about other cell cycle-related proteins (cyclin A, cyclin B, and cyclin E) were changed when MeCP2 was lost during cell cycle re-entry? Protein expression should be examined by western blot.

We appreciate your valuable suggestions. The expression of cell cycle related protein cyclin A2, cyclin B1 and cyclin E1 were evaluated by Western blotting. The expression of cyclin A2, cyclin B1 and cyclin E1 was enhanced by the knockdown of MeCP2 (Figure 4B). Conversely, the repressed expression of cyclin A2, cyclin B1 and cyclin E1 was observed by the over-expression of MeCP2 (Figure 4F).

(3) By combining MeCP2 ChIP-seq and RNA-seq of genes regulated by MeCP2, the authors uncovered the dual role of Mecp2 in preventing quiescence exit by targeting Rara and Nr1h3. All they show are the Q-PCR results. The authors should show the protein level of Rara and Nr1h3 when MeCP2 was lost during cell cycle re-entry.

Thank you for your advice. In Figure 7C, the knockdown efficiency of Rara and Nr1h3 were checked by Western blot analysis.

(4) The authors performed lentiviral and AAV-mediated gene knockdown to target Rara and Nr1h3 in Cells and Mecp2-cKO livers, respectively. The Knockdown efficacy should be verified by western blots (Fig 7 C and F).

In Figure 7F, the consequences of the Rara and Nr1h3 knockdown efficiency was verified by Western blot analysis.

(5) The other major concern is regarding the lack of quantitative assessments of MeCP2 WB results (Fig 2, Fig 4, and Fig 7).

Thank you for this suggestion. We added supplementary figures to Figure 2B, 2F and 2J to show the quantification membrane signal of MeCP2 protein in liver regeneration. And Fig S4A and 4B showing the quantification signal of MeCP2 protein in NIH3t3 cell cycle re-entry model.

(6) In the Figure legends of Fig 4 B and Fig 4F, the authors should delete the statistical descriptions, as there are no statistical results. In Fig 5F, Fig 5J, Fig 6D, Fig 7D and Fig7H, there are no statistical results of **p < 0.01, *p < 0.05 or ****p < 0.0001, respectively. The authors should check the description in the figure legends. In Fig S2C, the level of significance should be annotated.

We would like to express our heartfelt thanks for your thorough reading of our manuscript. We have made corrections to make manuscript clearer and more accurate. The level of significance have been annotated in Fig S2C.

(7) In Fig S4A, there are no WB results of Cyclin D1 and pRb, the authors should check the description.

Thank you for pointing this out. We have deleted the confusing statements in the revised manuscript.